# Di-BiLPS: Denoising induced Bidirectional Latent-PDE-Solver under Sparse Observations

Zhonghao Li[* 1]   Chaoyu Liu[* 2]   Qian Zhang[1]

## Abstract

Partial differential equations (PDEs) are fundamental for modeling complex natural and physical phenomena. In many real-world applications, however, observational data are **extremely sparse**, which severely limits the applicability of both classical numerical solvers and existing neural approaches. While neural methods have shown promising results under moderately sparse observations, their inference efficiency at high resolutions is limited, and their accuracy degrades substantially in the extremely sparse regime. In this work, we propose the **Di-BiLPS**, a unified neural framework that effectively handle **both forward and inverse** PDE problems under extremely sparse observations. Di-BiLPS combines a variational autoencoder to compress high-dimensional inputs into a compact latent space, a latent diffusion module to model uncertainty, and contrastive learning to align representations. Operating entirely in this latent space, the framework achieves efficient inference while retaining flexible input–output mapping. In addition, we introduce a **PDE-informed denoising algorithm** based on a variance-preserving diffusion process, which further improves inference efficiency. Extensive experiments on multiple PDE benchmarks demonstrate that Di-BiLPS consistently achieves **SOTA performance under extremely sparse inputs (as low as 3%)**, while substantially reducing computational cost. Moreover, Di-BiLPS enables **zero-shot super-resolution**, as it allows predictions over continuous spatial–temporal domains. Codes are

[*]Equal contribution   [1]School of Science, Harbin Institute of Technology, Shenzhen, China, lizhonghao@stu.hit.edu.cn [2]Department of Theoretical Physics and Applied Mathematics, University of Cambridge, Cambridge, United Kingdom, cl920@cam.ac.uk. Correspondence to: Qian Zhang <zhang.qian@hit.edu.cn>.

*Proceedings of the 43rd International Conference on Machine Learning*, Seoul, South Korea. PMLR 306, 2026. Copyright 2026 by the author(s).

## 1. Introduction

In the natural sciences, a wide range of physical, chemical, and biological processes can be modeled by partial differential equations (PDEs) (Roubíček, 2013; Wight & Zhao, 2020). In both academic research and industrial applications, numerical methods for PDEs are primarily concerned with simulating system states (the forward problem) and inferring unknown properties or parameters from experimental or observational data (the inverse problem).

Conventional numerical solvers are typically hand-crafted and restricted to specific classes of PDEs, requiring customization for each new formulation. This task-specific nature often results in limited generality, reduced flexibility, and significant computational overhead, especially when dealing with complex geometries or repeated simulations across varying conditions. To fill the gap, a number of learning-based approaches (Li et al., 2020b; 2024b) are proposed to learn the mappings between coefficient space and state space directly. These approaches differ mainly in how the solution operators are parameterized, leading to a variety of neural architectures, including graph-based methods (Li & Farimani, 2022), physics-informed networks (Raissi et al., 2019), and convolution-based models (Thuerey et al., 2020).

While neural PDE methods significantly improve generality and computational efficiency, they typically rely on access to dense observations of coefficients or system states to learn accurate solution operators. In real-world applications, however, such complete observations are rarely available. Instead, measurements are often extremely sparse across both spatial and temporal domains, which poses substantial challenges for accurate and reliable modeling. Under such extreme sparsity (e.g., observation rates below 3%), neither classical numerical solvers nor state-of-the-art neural PDE models (Wu et al., 2024; Hao et al., 2023) remain effective. This gap highlights the need for models that can robustly operate under severely limited observational data.

To this end, several diffusion-based (Huang et al., 2024;

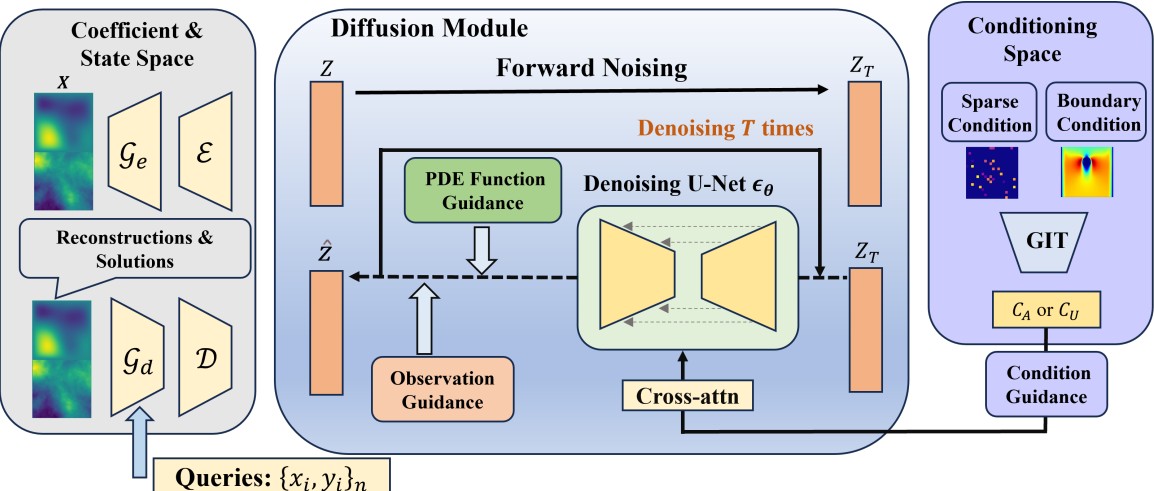

*Figure 1.* Overview of the architecture of **Di-BiLPS** : (a) Compression module via a pre-trained VAE (**left**), (b) Diffusion module with proposed PDE-Guided denoising algorithm(**middle**), and (c) Contrastive learning module (**right**). Proposed design leverages module (a) to extract informative and robust representations from extremely sparse inputs and performs high-speed denoising algorithm within the compressed latent space.

Shu et al., 2023) and graph-based (Zhao et al., 2022; Li et al., 2025b) methods have been proposed in recent years. However, these methods often suffer from high computational costs during inference, particularly when applied to densely discretized domains. Furthermore, for PDEs exhibiting drastic variations (Cen et al., 2024; Ma et al., 2024), interpolation-based methods are generally inadequate to reduce the reliance on dense discretization. To address this limitation, several latent neural PDE methods (Li et al., 2025a;c) have been proposed to learn solution mappings within compressed latent spaces.

Motivated by the aforementioned challenges, we propose *Di-BiLPS*, a unified and computationally efficient framework for learning and inference of PDEs under extreme data sparsity. As illustrated in Fig. 1, Di-BiLPS consists of three key components: (i) a contrastive learning module that aligns representations between sparse and full observations; (ii) a pre-trained variational autoencoder that encodes inputs into a compact latent space; and (iii) a latent diffusion model that enables bidirectional inference for PDE solutions within the latent domain.

Our main contributions can be summarized as follows:

(1) We propose *Di-BiLPS*, a scalable neural framework that jointly addresses forward and inverse PDE problems under extremely sparse observations. By reformulating PDE learning in a compressed latent space, Di-BiLPS achieves improved computational efficiency and enhanced input–output flexibility compared to existing methods.

(2) We develop a PDE-informed denoising algorithm

based on a variance-preserving diffusion process, which effectively integrates sparse observations with physical constraints to enable accurate bidirectional inference.

(3) Extensive experiments on five PDE benchmark datasets demonstrate that Di-BiLPS consistently outperforms state-of-the-art baselines in both prediction accuracy and computational efficiency. Benefitting from the nature of GINO, Di-BiLPS supports zero-shot resolution generalization, enabling inference on unseen spatial resolutions without retraining.

(4) We show that the proposed contrastive alignment and latent compression modules provide effective representations under extreme sparsity, suggesting their applicability to a broader class of latent neural PDE frameworks.

## 2. Related Work

**Diffusion informed Operator Learning:**  In the context of operator learning, diffusion models were first introduced for high-fidelity flow field reconstruction (Shu et al., 2023) and were subsequently extended to address ill-posed sparse prediction problems (Huang et al., 2024). In addition, governing-equation-driven denoising strategies have been explored to accelerate the solution of PDE-based diffusion models (Gao et al., 2024; Huang et al., 2024). ECI sampling(Cheng et al., 2025) also presents a novel zero-shot framework that adapts pre-trained flow-matching models to strictly satisfy physical hard constraints without gradient computations or fine-tuning for PDE-related scientific

generation tasks. Compared with other machine learning approaches, diffusion models exhibit an inherent ability to capture rich high-frequency components (Xu et al., 2025; Karumuri et al., 2026), making them particularly well suited for problems characterized by complex high-frequency dynamics, such as the Navier–Stokes equations(Molinaro et al., 2024). Further discussions on related work are provided in Appendix A.

## 3. Preliminaries

### 3.1. Problem Setup

Our study considers two categories of partial differential equations (PDEs): static PDEs and time-dependent(dynamic) PDEs. Static PDEs, such as Darcy flow and Poisson equations, describe equilibrium states of physical systems and are characterized by time-invariant parameters and solutions. These equations typically govern phenomena like fluid flow in porous media, heat conduction at steady state, and electrostatic potential distributions.

In contrast, time-dependent PDEs model the evolution of physical systems over time, such as Navier–Stokes equations for fluid dynamics. These PDEs incorporate temporal derivatives and require initial conditions in addition to boundary conditions. By addressing both static and dynamic PDEs, our framework aims to provide a unified solution approach capable of handling a broad class of forward and inverse problems arising in computational physics, engineering, and scientific machine learning.

Static systems are governed by time-independent coefficients and boundary conditions. They are typically defined by a function as follows:

$$F(P; A, U) = 0 \text{ in } \Omega \subset \mathbb{R}^d, U(P) = g(P) \text{ on } \partial\Omega. \quad (1)$$

where $\Omega$ is a bounded spatial domain, $P \in \Omega$ denotes a spatial coordinate, $A \in \mathcal{A}$ represents the PDE coefficient, and $U \in \mathcal{U}$ is the corresponding solution. The boundary of the domain is denoted by $\partial\Omega$, with the boundary condition given by $U|_{\partial\Omega} = g$. Our objective is to recover both the coefficient $A$ and the solution $U$ from sparse observations available on either $A$ or $U$.

Similarly, we consider dynamic systems, which can be formulated as follows:

$$F(P, t; A, U) = 0, \quad \text{in } \Omega \times (0, \infty), \quad (2)$$

$$U(P, t) = g(P, t), \quad \text{on } \partial\Omega \times (0, \infty),$$
$$U(P, t) = A, \quad \text{in } \overline{\Omega} \times \{0\}. \quad (3)$$

Here, $t$ denotes the temporal coordinate, $A := U(\cdot, 0) \in \mathcal{A}$ is the initial condition, and $U$ is the solution. The boundary constraint is given by $U|_{\partial\Omega \times (0,\infty)} = g$. Our objective is to

simultaneously recover both the initial condition $A$ and the solution at a specific time $t_i$, denoted by $U^{t_i} := U(\cdot, t_i) \in \mathcal{U}$, from sparse observations available on either $A$ or $U^{t_i}$. Here, we rewrite $U^{t_i}$ as $U$ for convenience in following parts since we don't care about $U$ at other time.

In our formulation, we denote the constraints $F$ defined in Eqs. (1) and (2) as the measurement condition $\mathcal{M} : F(\cdot) = 0$, while the initial and boundary conditions, $A$ and $g$, are collectively represented as the conditioning $C$.

## 4. Methodology

In this section, we present the compressed sensing module, the condition learning module and the latent diffusion model that constitute Di-BiLPS, as detailed in following subsections.

### 4.1. Variational Auto-Encoder for Irregular PDE Data

The successful application of Variational Autoencoders (VAEs) (Kingma et al., 2013) in Latent Diffusion Models (LDMs) (Rombach et al., 2022) has advanced the generation of high-resolution images. Performing machine learning in the compressed latent space not only accelerates the convergence of neural networks when fitting complex data, but also enhances their generalization capabilities.

In contrast to image processing, operator learning poses greater challenges in terms of data requirements. Models are required to handle inputs defined on irregular grids and perform prediction tasks over continuous solution spaces. To address these challenges, we propose a novel framework designed to meet both demands effectively.

Building upon the U-Net architecture (Ronneberger et al., 2015), we incorporate a GINO-based mesh encoder and decoder(see Appendix C) into the left and right sides of U-Net, respectively. Suppose the input data $V \in \mathbb{R}^{M \times (\dim(\Omega) + f)}$ has $f$ features defined in domain $\Omega$. In our setting, we compress $\mathcal{A}$ and $\mathcal{U}$ jointly in order to model the underlying interactions. As shown in Eq. (4), through the GINO-based mesh encoder, irregular input data $V$ is first embedded into a structured $w$-resolution grid representation $\mathcal{G}_e(V) \in \mathbb{R}^{h \times w^{\dim(\Omega)}}$, enabling it to be processed by the encoder $\mathcal{E}$ of U-Net. Afterward, we obtain the processed latent embeddings $Z \in \mathbb{R}^{l \times w_l^{\dim(\Omega)}}$ with $l$ latent channels and $w_l$ resolution. Subsequently, for a given query point $q \in \Omega$, the GINO-based decoder $\mathcal{G}_d$ infers the solution based on neighboring points of decoded embeddings $\mathcal{D}(Z)$.

$$Z = \mathcal{E}(\mathcal{G}_e(V)) = \text{Unet}_{\text{L}}\big(\text{GINOEnc}(V)\big),$$
$$v_q = \mathcal{G}_d(q, \mathcal{D}(Z)) = \text{GINODec}\big(q, \text{Unet}_{\text{R}}(Z)\big), \quad (4)$$

where $\text{Unet}_{\text{L}}$ and $\text{Unet}_{\text{R}}$ are the down-convolution and up-convolution parts of U-net, respectively.

**Algorithm 1** Training Algorithm

**Input**: VAE-Encoder $\mathcal{E}, \mathcal{G}_e$, NumSample $N$
LearnedCondition $C_A$ and $C_U$, JointInput $V$, DenoisingSampler $S$, MaxTimeStep $\tau$

1: $\{\alpha_0, \cdots, \alpha_\tau\} \longleftarrow S$
2: **repeat**
3:     $i \sim \text{Uniform}(\{1, \cdots, N\})$
4:     $t \sim \text{Uniform}(\{1, \cdots, \tau\})$
5:     $c \sim \text{Uniform}(\{c_{iA}, c_{iU}\})$
6:     $Z_i = \mathcal{E}(\mathcal{G}_e(V_i))$
7:     Sample $\epsilon \sim N(0, I)$
8:     $Z_t = \sqrt{\alpha_t} Z_i + \sqrt{1 - \alpha_t} \epsilon$
9:     Taking gradient descent step on
       $\nabla_\theta \left\| \epsilon_\theta^{(t)}(Z_t, c) - \epsilon \right\|_2^2$
10:    Update parameters $\theta$
11: **until** converged

---

Let $V_p$ denote the positional coordinates of the input data $V$. The restructured data $\hat{V}$ at the original input points $V_p$ can be expressed as follows:

$$\hat{V} = \mathcal{G}_d(V_p, \mathcal{D}(Z)). \tag{5}$$

Here, MSE loss, Perceptual loss (Johnson et al., 2016) and KL loss (Kingma et al., 2013) are adopted to train the GiT as follows:

$$Loss = \text{MSE}(V, \hat{V}) + \text{PercLoss}(V, \hat{V}) + \text{KL}(Z). \tag{6}$$

The design of our VAE is also shown in Fig. 1 and the detailed implementation of GINO (Li et al., 2023) is deferred to Sec. C.

### 4.2. Condition Learning under Sparse Observations

In the operator learning benchmarks (Boullé & Townsend, 2024), neural PDE methods are designed to solve the forward process ($\mathcal{F} : \mathcal{A} \to \mathcal{U}$) and the inverse process ($\mathcal{I} : \mathcal{U} \to \mathcal{A}$) separately. In practice, samples in $\mathcal{A}$ is discretized as a set $\{A_i\}$ defined over either a structured or irregular grid, with the grid points sampled from domain $\Omega$. The same discretization strategy is applied to samples in $\mathcal{U}$, yielding $U_i$. Although finer discretizations encode more detailed information, much of it may be irrelevant to the objective mappings $\mathcal{F}$ and $\mathcal{I}$.

In many practical scenarios, due to limitations such as instrument precision, we can typically observe only partial measurements of physical properties $\mathcal{A}$ or system states $\mathcal{U}$. Since $\mathcal{U}$ and $\mathcal{A}$ lie in a continuous function spaces, we believe that appropriately sparse partial observations over a discretized grid can still retain most of the interactions relevant to the underlying mappings $\mathcal{F}$ and $\mathcal{I}$. To formalize this, we define $\mathcal{P}^A$ and $\mathcal{P}^U$ as the sparse subspaces of $\mathcal{A}$

and $\mathcal{U}$, respectively.

To effectively capture interactions, we propose a contrastive learning framework with a novel encoder. As illustrated in Fig. 2, the encoder integrates GINO-based mesh encoder (Li et al., 2023) with a Vision Transformer (ViT) encoder (Dosovitskiy et al., 2020). Suppose $P_i^A$ and $P_i^U$ are sparse point clouds, downsampled from the discretizations $A_i$ and $U_i$[1], respectively. Within the proposed contrastive learning framework, our goal is to learn the interactions between the sparse observations and their corresponding original spaces. In fact, we seek to improve the effectiveness of learning the mappings $\mathcal{F}^S : \mathcal{P}^A \to \mathcal{U}$ and $\mathcal{I}^S : \mathcal{P}^U \to \mathcal{A}$.

Let $A_i$ and $U_i$ denote the discretized representations in $\mathcal{A}$ and $\mathcal{U}$, respectively. Correspondingly, the sparse observations $P_i^A \in \mathbb{R}^{M \times (\dim(\Omega) + f_a)}$ and $P_i^U \in \mathbb{R}^{M \times (\dim(\Omega) + f_u)}$ consist of $M$ points sampled from $A_i$ and $U_i$, respectively. $f_a$ and $f_u$ are the feature quantity of $A$ and $U$. The associated boundary condition is denoted as $B_i$. By applying the encoder defined in Eq. (7), the $i$-th sparse observations $P_i^j$ are encoded into an $h$-dim conditioning space $c_{ij} \in \mathbb{R}^h$, while the full observations $A_i$ and $U_i$ are encoded into an $h$-dim key spaces $k_{iA} \in \mathbb{R}^h$ and $k_{iU} \in \mathbb{R}^h$, respectively.

$$
\begin{aligned}
c_{ij} &= \text{ViT}_j \left[ \text{GINOEnc}_j(P_i^j), B_i \right], \quad j = A, U, \\
k_{iA} &= \text{ViT}_A \left[ \text{ResNet}_A(A_i), B_i \right], \\
k_{iU} &= \text{ViT}_U \left[ \text{ResNet}_U(U_i), B_i \right],
\end{aligned} \tag{7}
$$

where modules $\text{GINOEnc}_A$ and $\text{ResNet}_A$ embed $P_i^A$ and $A_i$ into a shared structured grid format, thereby enabling their outputs to be processed by a shared encoder $\text{ViT}_A$. The same procedure is applied to $U_i$.

Let $C_A$, $C_U$, $K_A$, and $K_U \in \mathbb{R}^{N \times h}$ denote the matrices formed by stacking and normalizing the vectors $c_{iA}$, $c_{iU}$, $k_{iA}$, and $k_{iU}$ as rows, respectively. Here $N$ is the batch size.

To capture the underlying interactions between $C$ and $K$, we compute the pairwise cosine similarity matrix $CK^T$ as matching scores, as described in Eq. (8). In our contrastive learning setup, matching scores $S^F$ and $S^I$ along the main diagonal represent the similarity between matched pairs, reflecting how well each condition in $C$ aligns with its corresponding key in $K$.

$$S^F = C_A K_U^T, \quad S^I = C_U K_A^T, \tag{8}$$

where $K_U^T$ and $K_A^T$ are the transposes of $K_U$ and $K_A$ respectively. The loss function can be expressed as follows:

$$
\begin{aligned}
Loss = &\text{CE}(S^F, I) + \text{CE}(S^I, I) + \\
&\text{CE}\left((S^F)^T, I\right) + \text{CE}\left((S^I)^T, I\right),
\end{aligned} \tag{9}
$$

where CE is the cross-entropy loss function and $I \in \mathbb{R}^{N \times N}$

---

[1] i.e., Sampled from subspaces $\mathcal{P}^A$ and $\mathcal{P}^U$, respectively.

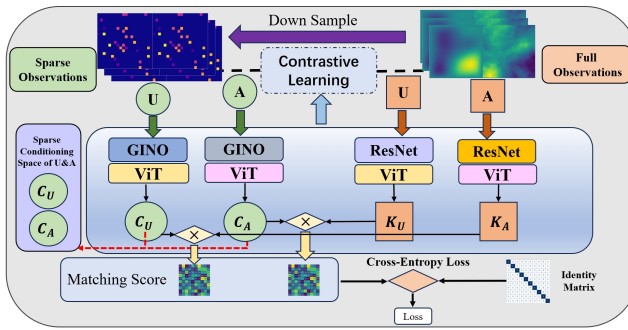

*Figure 2.* Illustration of contrastive learning module. Proposed GINO-ViT framework encodes both sparse and full observations into a unified latent space. By optimizing the match score within this space, framework is designed to capture informative and robust representations between sparse and full observations. Here, we emphasize that ViT modules with the same color **share weights** in this figure.

is the identity matrix. The pseudocode for the core of an implementation of GiT is shown in D.

After training GiT, we obtain the conditioning embeddings $C_A$ and $C_U$ from the sparse observations $P^A$ and $P^U$. These embeddings facilitate more efficient mappings $\mathcal{C}_{U,A} \to \mathcal{U}, \mathcal{A}$, compared to the original mappings $\mathcal{F}^S, \mathcal{I}^S$.

### 4.3. PDE-Guided Denoising Algorithm for Diffusion Models

Diffusion models comprise a predefined forward process that gradually corrupts the data by adding Gaussian noise, and a learnable reverse process that iteratively denoises the noisy inputs to recover the original data distribution. As described in DDIM (Song et al., 2020a), the mathematical formulations of both the forward and reverse processes are given as follows:

$$q(Z_t|Z_0) = \mathcal{N}(\sqrt{\alpha_t}Z_0, (1-\alpha_t)I), \quad (10)$$

$$q(Z_{t-1}|Z_t, Z_0) = \mathcal{N}(\sqrt{\alpha_{t-1}}Z_0 + \frac{\sqrt{1-\alpha_{t-1}-\sigma_t^2}(Z_t - \sqrt{\alpha_t}Z_0)}{\sqrt{1-\alpha_t}}, \sigma_t^2 I). \quad (11)$$

Here, we consider the case where $\sigma_t = 0$, corresponding to a deterministic denoising process. According to Eq. (11), the latent variable $Z_{t-1}$ can be obtained from $Z_t$ as follows:

$$Z_{t-1} = \sqrt{\alpha_{t-1}}Z_0 + \sqrt{1-\alpha_{t-1}}\epsilon_\theta^{(t)}(Z_t), \quad (12)$$

where $Z_0$ can be reconstructed from $Z_t$ via $Z_0 = \frac{Z_t - \sqrt{1-\alpha_t}\epsilon_\theta^{(t)}(Z_t)}{\sqrt{\alpha_t}}$. Under the framework of stochastic differential equations (SDEs) (Song et al., 2020b), Eq. (12) can be reformulated by leveraging the identity $\nabla_{Z_t} \log p_\theta(Z_t) = -\frac{\epsilon_\theta^{(t)}(Z_t)}{\sqrt{1-\alpha_t}}$. To facilitate controllable denoising, classifier-free guided diffusion models (Liu et al., 2023) design a

noise estimation network based on conditional probabilistic formulation $p_\theta(Z_t|C) = -\frac{\epsilon_\theta^{(t)}(Z_t, C)}{\sqrt{1-\alpha_t}}$. In the context of our

---

**Algorithm 2** Guided Diffusion Sampling Algorithm

**Input**: GuidedDiffusionModel $\epsilon_\theta$, VAE-Decoder $\mathcal{D}, \mathcal{G}_d$, LearnedCondition $C$, PDEConstraint $\mathcal{M}, Q_m$, Observations $P = \{P_p, P_v\}$, Weights $\zeta_{obs}, \zeta_{pde}$, DenoisingSampler $S$, NumTimestep $T$, Query positions: $Q$,
**Output**: $X$

1: $\{t_0, \cdots, t_T\}, \{\alpha_{t_0}, \cdots, \alpha_{t_T}\} \longleftarrow S(T)$
   // Get $\alpha$ from sampler
2: Sample $Z_{t_T}$ from $\mathcal{N}(0, I)$
3: **for** $i = T$ to 1 **do**
4:   $\alpha_i \leftarrow \alpha_{t_i}, \alpha_{i-1} \leftarrow \alpha_{t_{i-1}}$
     // Get variance at $t$
5:   $N_{cond} \leftarrow \epsilon_\theta^{(t_i)}(Z_{t_i}, C)$
     // Estimate conditional noise
6:   $\hat{Z}_0 \leftarrow \frac{1}{\sqrt{\alpha_i}}(Z_{t_i} - \sqrt{1-\alpha_i}N_{cond})$
     // Estimate $\hat{Z}_0$
7:   $\hat{X}_P \leftarrow \mathcal{G}_d\left(P_p, \mathcal{D}(\hat{Z}_0)\right)$
     // Decode latent embeddings at observed positions.
8:   $\hat{X}_F \leftarrow \mathcal{G}_d\left(Q_m, \mathcal{D}(\hat{Z}_0)\right)$
     // Decode latent embeddings at grid positions.
9:   $N_{pde} \leftarrow -\zeta_{pde}\nabla_{Z_{t_i}}\left\|F(\hat{X}_F)\right\|_2^2$
     // Calculate Eq. (15)
10:  $N_{obs} \leftarrow -\frac{\zeta_{obs}}{M}\nabla_{\hat{Z}_0}\left\|\hat{X}_P - P_v\right\|_2^2$
     // Calculate Eq. (17)
11:  $S \leftarrow -\frac{N_{cond}}{\sqrt{1-\alpha_i}} - N_{pde}$
     // Calculate Eq. (16)
12:  $Z_{t_{i-1}} \leftarrow \frac{\sqrt{\alpha_{i-1}}}{\sqrt{\alpha_i}}Z_{t_i} - \left(\sqrt{\frac{1-\alpha_{i-1}}{\alpha_{i-1}}} - \sqrt{\frac{1-\alpha_i}{\alpha_i}}\right)\sqrt{\alpha_{i-1}(1-\alpha_i)}S$
     // Do denoising step as Eq. (13)
13:  $Z_{t_{i-1}} \leftarrow Z_{t_{i-1}} + N_{obs}$
     // Guidance to Sparse Observations
14: **end for**
15: $X \leftarrow \mathcal{G}_d\left(\mathcal{D}(Z_0), Q\right)$   // $t_0 = 0$
    // Decode latent embeddings on query positions.
16: **return** $X$

---

problem, we employ the guided diffusion approach with the formulation $p_\theta(Z_t \mid C, \mathcal{M})$, where $C$ and $\mathcal{M}$ represent the conditioning information defined in Sec. 3.1. Under the assumptions of Eqs, (10) and (11), the accelerated sampling method proposed by DDIM (Song et al., 2020a) can be formulated as follows:

$$\frac{Z_\tau}{\sqrt{\alpha_\tau}} = \frac{Z_t}{\sqrt{\alpha_t}} - \left(\sqrt{\frac{1-\alpha_\tau}{\alpha_\tau}} - \sqrt{\frac{1-\alpha_t}{\alpha_t}}\right)\sqrt{1-\alpha_t} \nabla_{Z_t}\log p_\theta(Z_t|C, \mathcal{M}), \quad (13)$$

where $\tau \in \{0, 1, \cdots, t-1\}$ denotes a previous timestep. Since the condition $\mathcal{M}$ acts as a measurement operator $F(\cdot)$ applied to the embeddings $\mathcal{D}(Z_0)$, the conditional probability can be decomposed into the following components:

$$
\begin{aligned}
p_\theta(Z_t|C, \mathcal{M}) &\propto p_\theta(Z_t|C) * p_\theta(\mathcal{M}|Z_t, C) \\
&= p_\theta(Z_t|C) * p_\theta(\mathcal{M}|Z_0(Z_t, C)).
\end{aligned}
\tag{14}
$$

According to DPS (Chung et al., 2022) and diffusion-PDE (Huang et al., 2024), if measurement operator $F(\cdot)$ is corrupted by Gaussian noise with some standard deviation $\delta$ (i.e., $\mathcal{M}|Z_0 \sim \mathcal{N}(F(\mathcal{D}(Z_0)), \delta^2 I)$), then the log-likelihood function can be approximated accordingly as follows:

$$
\begin{aligned}
N_{pde} &= \nabla_{Z_t} \log p_\theta\big(\mathcal{M}|Z_0(Z_t, C)\big) \\
&\approx \nabla_{Z_t} \log p_\theta\big(\mathcal{M}|\hat{Z}_0(Z_t, C)\big) \\
&\approx -\frac{1}{\delta^2} \nabla_{Z_t} \left\| F\left(\mathcal{G}_d[Q_m, \mathcal{D}(\hat{Z}_0(Z_t, C))]\right) \right\|_2^2,
\end{aligned}
\tag{15}
$$

where $Q_m$ typically denotes a discretized grid over the domain $\Omega$, enabling the computation of $F$. Here, we rewrite $\frac{1}{\delta^2}$ as $\zeta_{pde}$ for convenience. To summarize Eqs. (14) and (15), the $\nabla_{Z_t} \log p_\theta(Z_t \mid C, \mathcal{M})$ can be estimated as follows:

$$
\begin{aligned}
S &:= \nabla_{Z_t} \log p_\theta(Z_t|C, \mathcal{M}) \\
&= \nabla_{Z_t} \log p_\theta(Z_t|C) + \nabla_{Z_t} \log p_\theta\big(\mathcal{M}|Z_0(Z_t, C)\big) \\
&\approx -\frac{\epsilon_\theta^{(t)}(Z_t, C)}{\sqrt{1-\alpha_t}} - \zeta_{pde} \nabla_{Z_t} \left\| F\left(\mathcal{G}_d[Q_m, \mathcal{D}(\hat{Z}_0)]\right) \right\|_2^2,
\end{aligned}
\tag{16}
$$

where $\hat{Z}_0(Z_t, C) = \frac{Z_t - \sqrt{1-\alpha_t}\epsilon_\theta^{(t)}(Z_t, C)}{\sqrt{\alpha_t}}$. In addition, since the latent variable $\hat{Z}_0$ has been estimated at each denoising step, we introduce an additional $\ell_2$ penalty term that measures the discrepancy between the observed points $P = \{P_p, P_v\}$ and their corresponding reconstructions in $\hat{Z}_0$. As shown in Eq. (17), this penalty is incorporated into the denoising process to further guide $\hat{Z}_0(Z_t, C)$ toward sparse observations.

$$
N_{obs} = -\frac{\zeta_{obs}}{M} \nabla_{\hat{Z}_0} \left\| \mathcal{G}_d(\mathcal{D}(\hat{Z}_0(Z_t, C), P_p)) - P_v \right\|_2^2, \tag{17}
$$

where $P_v$ is the function value at the observed points $P_p$.

### 4.4. Architecture of Diffusion Model

As discussed in Sec. 4.2, our approach enables more efficient mappings from $\mathcal{C}^A$ to $\mathcal{U}$ and from $\mathcal{C}^U$ to $\mathcal{A}$, compared to the original mappings $\mathcal{F}^S$ and $\mathcal{I}^S$. In many practical scenarios, however, full observations are unavailable due to measurement limitations, making sparse recovery in the observation space equally important. To address this, we propose a novel latent diffusion model that learns the joint mappings $\mathcal{C}^A \rightarrow \{\mathcal{U}, \mathcal{A}\}$ and $\mathcal{C}^U \rightarrow \{\mathcal{U}, \mathcal{A}\}$ within a unified generative framework as shown in (Fig. 1).

Here, we aim to model the joint distribution over $\mathcal{X} = \mathcal{A} \times \mathcal{U}$ using VAE. Given $i$-th joint input observation $V_i \in \mathbb{R}^{M \times (\dim(\Omega)+f)}$ in irregular grids[2], where $f = f_a + f_u$, the VAE encodes $V_i$ into a latent representation $Z_i$, as illustrated in Eq. (4). Therefore, regardless of whether the diffusion module is conditioned on $c_{iA}$ or $c_{iU}$ to predict the latent representation $Z_i$, both $A_i$ and $U_i$ can be subsequently reconstructed from $Z_i$.

Our noise estimation network $\epsilon_\theta$ is built upon the U-Net architecture (Ronneberger et al., 2015). Given an input sample $V_i$ and a randomly selected timestep $t$, the training procedure is summarized in Algorithm 1, where the conditioning embeddings $C_A$ and $C_U$ are precomputed using the model described in Eq. (7).

Once the noise estimation network $\epsilon_\theta$ has been trained, it can be used to perform inference on any point set $Q$ within the domain $\Omega$. The detailed inference procedure is outlined in Algorithm 2. We emphasize that $P_p$ and $P_v$ denote the positions and values of the sparse observations. In the forward and inverse problems, sparse observations $P = \{P_p, P_v\}$ are defined over $\mathcal{A}$ and $\mathcal{U}$, respectively. Accordingly, the way $C$ is precomputed from $P$ in Eq. (7) also depends on the space of $P$ in Algorithm 2.

## 5. Experiments

In this section, we evaluate Di-BiLPS on five PDE benchmarks under sparse observational settings, covering both forward and inverse operator learning tasks. Results demonstrate consistent improvements over existing operator learning and diffusion-based baselines in terms of accuracy and efficiency.

### 5.1. Experiment Settings

**Dataset Preparation:** In our experimental setup, PDEs are formulated on the unit square domain $\Omega = (0,1)^2$, represented by a uniform $128 \times 128$ grid. The training dataset is obtained through numerical simulation using the Finite Element Method (FEM). To enable fair and consistent comparison with prior work, our pipeline Di-BiLPS: is evaluated on five standard PDE benchmark datasets(Darcy Flow, Helmholtz, Poisson, Bounded Navier-Stokes, and Non-Bounded Navier-Stokes) proposed by DiffusionPDE (Huang et al., 2024) and FNO (Li et al., 2020b). Detailed information of datasets above is shown in B.1.

**Baseline Models:** We evaluate Di-BiLPS by comparing it against both operator learning baselines and representative models on uncertain prediction in operator learning. For

---

[2]Without loss of generality, if $V_i \in \mathbb{R}^{f \times w^{\dim(\Omega)}}$, we employ a ResNet-based encoder $\mathcal{G}_e$ to map $V_i$ into a latent representation of the same shape as GINO.

*Table 1.* Comparison of the relative $\ell_2$ error of various methods on forward and inverse PDE tasks. The runtime performance (in seconds) of our model and DiffusionPDE is also reported. The best results in each task are highlighted in **bold**.

| Task | PDE Func | Ours($\ell_2$ \| RunTime) | | DiffPDE | PINO | DeepONet | PINNs | FNO |
|------|----------|------|------|---------|------|----------|-------|-----|
| Forward | Darcy Flow | **0.021**(↓16%) | **35.45**(↓86%) | 0.025 \| 255.74 | 0.352 | 0.383 | 0.488 | 0.282 |
| | Poisson | **0.038**(↓16%) | **19.69**(↓92%) | 0.045 \| 255.37 | 1.071 | 1.555 | 1.281 | 1.009 |
| | Helmholtz | **0.025**(↓72%) | **20.52**(↓92%) | 0.088 \| 249.81 | 1.065 | 1.231 | 1.423 | 0.982 |
| | Non-B NS | **0.043**(↓38%) | **19.82**(↓92%) | 0.069 \| 256.40 | 1.014 | 1.032 | 1.427 | 1.014 |
| | Bounded NS | **0.021**(↓46%) | **19.08**(↓93%) | 0.039 \| 256.54 | 0.811 | 0.977 | 1.001 | 0.828 |
| Inverse | Darcy Flow | **0.013**(↓59%) | **36.04**(↓86%) | 0.032 \| 255.74 | 0.492 | 0.411 | 0.597 | 0.493 |
| | Poisson | **0.123**(↓38%) | **19.59**(↓92%) | 0.200 \| 254.22 | 2.319 | 1.058 | 1.300 | 2.327 |
| | Helmholtz | **0.120**(↓47%) | **20.56**(↓92%) | 0.226 \| 252.72 | 2.169 | 1.328 | 1.600 | 2.182 |
| | Non-B NS | 0.116(↑12%) | **19.52**(↓92%) | **0.104** \| 254.18 | 0.960 | 0.972 | 1.468 | 0.960 |
| | Bounded NS | 0.032(↑19%) | **19.80**(↓92%) | **0.027** \| 253.41 | 0.695 | 0.919 | 1.055 | 0.696 |

operator learning, we consider PINO (Li et al., 2024b), DeepONet (Lu et al., 2021), PINNs (Raissi et al., 2019), and FNO (Li et al., 2020b). For uncertain prediction in operator learning, we include DiffusionPDE (Huang et al., 2024) and GraphPDE (Zhao et al., 2022) as comparative baselines. We evaluate Di-BiLPS on five benchmark PDE datasets under sparse observational settings, covering both forward and inverse operator learning tasks.

**Metrics:** In our experiments, we use the relative $\ell_2$ error to evaluate model performance, defined as $\ell_2(x, \hat{x}) = \frac{\|x - \hat{x}\|_2}{\|x\|_2}$, where $x$ is the ground truth and $\hat{x}$ is the prediction.

**Setup:** We conducted our experiments within an open source framework *diffusers*(von Platen et al., 2022) in huggingface. All experiments were run on 8 Nvidia RTX4090 GPUs with 24GB memory.

### 5.2. Training Pipeline

In the standard operator learning benchmark, paired data $(A_i, U_i)$ is provided to learn the mappings $A_i \leftrightarrow U_i$. In our setting, however, only 500 points(3%) in $A_i$ or $U_i$ are sampled to learn these bidirectional mappings. First, as outlined in Sec. 4.2, we train a condition learning module(GiT) to extract robust message from 3% available observations. Second, we train a GINO-VAE to encode the paired data into a latent space, with a compression rate of 6.25%. Third, the noise estimation network $\epsilon_\theta$ is trained using compressed latent representations from the GINO-VAE, together with corresponding conditioning information derived from the trained conditional learning module.

### 5.3. Main Evaluation Results

Table 1 presents the results of our model and other baseline models over 10 runs. Since the coefficients of Darcy Flow are binary, we evaluate the error rates of our predictions. Across five PDE benchmark datasets, our model consis-

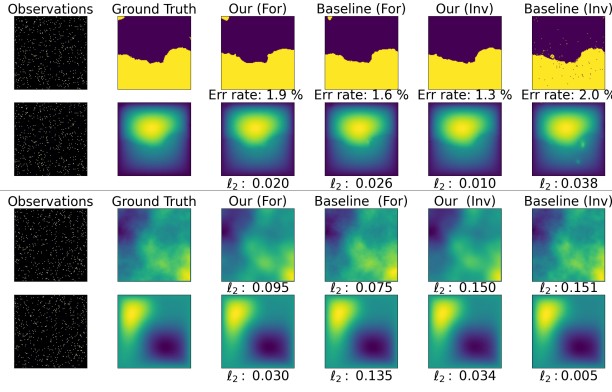

*Figure 3.* Comparison between our model and DiffusionPDE on the Darcy Flow (**top**) and inhomogeneous Helmholtz equation (**bottom**) tasks.

tently outperforms existing operator learning approaches. It surpasses the diffusion-based approach DiffusionPDE on 8 out of 10 tasks, while achieving a substantial reduction in prediction error and a 90% decrease in inference time, highlighting both its accuracy and efficiency. In the inverse problems of the Poisson and Helmholtz equations, the performance of DiffusionPDE deteriorates due to inadequate constraints in the coefficient space arising from randomly generated fields. Our model addresses this limitation effectively by incorporating the proposed condition learning module, resulting a 47% error reduction. Throughout all tasks, the GPU memory consumption of our model was 8.6 GB, compared to 4.5 GB for DiffusionPDE.

Fig. 3 presents a comprehensive comparison between our model and DiffusionPDE, covering both forward and inverse problems on the Darcy Flow and Helmholtz equations. The results in Fig. 3 indicate that our model achieves superior performance in predictive accuracy, whereas DiffusionPDE demonstrates a stronger ability to recover the original coefficient space.

*Table 2.* Relative $\ell_2$ error for ablation study on Di-BiLPS. This table illustrates the effect of removing key components in both forward and inverse PDE tasks. The best results in each task are highlighted in **bold**.

| Method | Darcy Flow | | Poisson | | Helmholtz | | Non-Bounded NS | | Bounded NS | |
| --- | --- | --- | --- | --- | --- | --- | --- | --- | --- | --- |
| Direction | For | Inv | For | Inv | For | Inv | For | Inv | For | Inv |
| Di-BiLPS | **0.021** | **0.013** | **0.038** | **0.123** | **0.025** | **0.120** | **0.043** | **0.116** | **0.021** | **0.032** |
| w/o *PDE Guidance* | 0.021 | 0.013 | 0.038 | 0.123 | 0.026 | 0.120 | 0.044 | 0.116 | 0.021 | 0.032 |
| w/o *Observation Guidance* | 0.042 | 0.040 | 0.108 | 0.450 | 0.140 | 0.339 | 0.274 | 0.414 | 0.109 | 0.074 |
| w/o *Condition Guidance* | 0.237 | 0.272 | 1.185 | 0.292 | 0.573 | 1.288 | 0.790 | 1.429 | 0.189 | 0.978 |

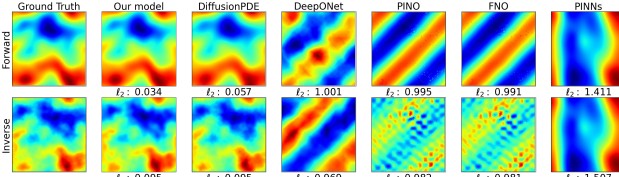

*Figure 4.* Comparison of all baseline models(Huang et al., 2024) listed in Table 1 on bidirectional problems of Non-bounded Navier–Stokes equations.

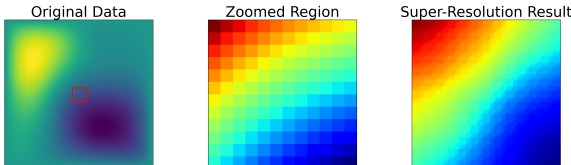

*Figure 6.* Zero-shot super-resolution results on the forward problem of the inhomogeneous Helmholtz equations.

Fig. 4 presents a comprehensive comparison of all baseline models listed in Table 1, encompassing both forward and inverse problems for the non-bounded Navier–Stokes equations. The results indicate that most existing methods are unable to effectively solve PDEs under sparse observational settings. In contrast, our model achieves accurate and efficient predictions.

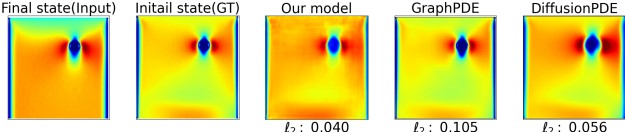

*Figure 5.* Comparison among our model, GraphPDE and DiffusionPDE(Huang et al., 2024) on the inverse problem of Bounded Navier-Stokes equations.

Fig. 5 presents a comparison among our model, GraphPDE and DiffusionPDE on the inverse problem of the bounded Navier-Stokes equation. The results in Fig. 5 indicate that our model more accurately captures the dynamical states around the cylinder, resulting in a lower overall prediction error.

We also conducted additional experiments to explore the generalization of sparsity settings. We retrained the contrastive learning module at 1% and 10% observation densities with the encoder of full observations frozen, and reused VAE and diffusion modules. As shown in Appendix G, the performance degrades gracefully as sparsity increases, indicating that our method remains stable across a range of observation regimes and easy to generalize to different sparsity regimes.

### 5.4. Ablation Studies

As shown in Table 2, we conduct an ablation study to evaluate the contribution of each component in our model across five benchmark PDEs in both forward and inverse settings. In this table, *PDE Guidance* corresponds to the term $N_{pde}$ as defined in Eq. (15), *Observation Guidance* refers to $N_{obs}$ defined in Eq. (17), and *Condition Guidance* denotes the conditional input $C$ to the noise estimation network $\epsilon_\theta$. Removing the PDE guidance results in negligible performance degradation, suggesting its auxiliary role. However, excluding the observation guidance leads to a significant increase in relative $\ell_2$ error across all tasks, highlighting its critical role in leveraging sparse observations. Furthermore, removing the condition guidance causes severe performance deterioration, particularly in complex scenarios such as the Non-Bounded Navier–Stokes equations, indicating its essential role in capturing contextual dependencies.

### 5.5. Zero-shot Super-Resolution

As discussed in Sec. 4.1, our VAE model incorporates the encoder and decoder from GINO. As a result, our model is inherently capable of performing predictions over a continuous spatial domain. This property enables seamless super-resolution inference in the solution space by querying the decoder of GINO at arbitrary spatial locations. Notably, although the training data contains only low-resolution samples, our model can generate high-resolution predictions in a zero-shot manner by specifying query points at a finer resolution. Fig. 6 illustrates the zero-shot super-resolution results on the forward inhomogeneous Helmholtz equations.

## 6. Limitations

Despite these encouraging results, several aspects of Di-BiLPS deserve further discussion. First, within the scope of diffusion-informed operator learning, developing `governing equation-driven` denoising algorithms is a vital paradigm for incorporating physical priors. However, our ablation studies demonstrate that directly performing gradient descent with the loss derived from governing equations(i.e., PDE guidance) barely improves model performance, especially for diffusion models in the latent space. Our current implementation of operator learning tailored for latent diffusion models still has certain limitations.

Second, our model deteriorates drastically when the observation rate decreases to 0.1%. Exploring the minimal sparsity threshold for valid model performance remains a valuable avenue for future work.

## 7. Conclusion

In this work, we propose Di-BiLPS, a unified and effective neural framework for solving bidirectional PDE problems under highly sparse settings. Our results highlight the application potential of latent PDE solvers through leveraging carefully constructed low-rank representations. Moreover, our findings in this work demonstrate that diffusion-based models exhibit robust predictive performance in sparse data scenarios of industrial applications by directly learning the underlying distribution of data. We anticipate that both Di-BiLPS and the corresponding proposed PDE denoising algorithm will offer valuable insights and serve as a foundation for future research on more complex PDE challenges.

## Acknowledgement

The authors would like to thank the editor and the anonymous reviewers for their valuable comments and constructive suggestions, which have helped improve the quality of this manuscript. The authors are also grateful to Professor Zhonghua Qiao from The Hong Kong Polytechnic University and Professor Dong Wang from The Chinese University of Hong Kong, Shenzhen, for their helpful discussions and support. Qian Zhang acknowledges the support from the National Natural Science Foundation of China (Grant No. 12401480) and the Shenzhen Talent Start-up Fund (Grant No. ZX2023295). Chaoyu Liu acknowledges the support from the Maths4DL program under grant EP/V026259/1.

## Impact Statement

This paper presents work whose goal is to advance the field of Machine Learning by developing more efficient and flexible learning-based methods for scientific computing problems governed by partial differential equations. The techniques introduced in this work are methodological in nature and are evaluated on standard benchmark datasets. By improving computational efficiency and robustness under limited observations, such methods may support more efficient modeling and inference in real-world scientific and engineering systems. We do not anticipate any direct negative societal or ethical consequences arising from this work.

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

# A. Related Work

## A.1. Operator Learning

Operator learning aims to approximate mappings between infinite dimensional function spaces, typically from coefficients, source terms, or initial conditions to the corresponding solution functions. Instead of explicitly solving the underlying partial differential equations, these methods train neural networks to learn such functional correspondences directly from data. A pioneering framework in this direction is DeepONet (Lu et al., 2021; Goswami et al., 2022), which establishes the feasibility of approximating nonlinear operators using neural networks. Subsequent research has developed a variety of neural operators based on kernel integral formulations (Li et al., 2020a;b; Kovachki et al., 2023). Within this family, the Galerkin Transformer (Cao, 2021) introduces an attention mechanism inspired by Galerkin projections to realize kernel integral operators. Building upon this idea, OFormer (Li et al., 2022) separates observation and query locations through a cross-attention design, enabling more flexible operator approximation. GNOT (Hao et al., 2023) further enhances adaptability to heterogeneous inputs by employing normalized cross-attention and reduces computational overhead through a linear-time attention mechanism. FactFormer (Li et al., 2024a) factorizes kernel integrals along axial directions, decomposing multi-dimensional functions into multiple 1D components, thereby lowering computational complexity. ONO (Xiao et al., 2023) improves generalization by incorporating orthogonality regularization into its attention layers.

Another line of work focuses on constructing operators in latent spaces. Transolver (Wu et al., 2024) alternates between geometric and physical representations using physics-attention, enabling expressive latent modeling. LNO(Wang & Wang, 2024) adopts an encoder-decoder architecture built upon physics-attention to tackle heterogeneous geometric problems. LSM (Wu et al., 2023) encodes input functions via cross-attention into a latent space where a set of orthogonal bases is learned, and then reconstructs outputs back to the geometric domain via a second cross-attention module. UPT (Alkin et al., 2024) introduces additional encoding and decoding objectives and compresses geometric information into super-nodes with graph neural networks (Scarselli et al., 2008), enabling more compact latent representations.

In contrast to these approaches, our method learns the bidirectional mapping through latent representations in a fully end-to-end manner. Leveraging cross-attention, our framework avoids hand-crafted feature construction or auxiliary loss terms, while effectively capturing heterogeneous functional interactions.

## A.2. Latent Diffusion Models

Latent Diffusion Models(LDMs) (Rombach et al., 2022), introduced in 2021, have rapidly emerged as a transformative framework within the field of generative modeling. By performing the diffusion process in a compact latent space rather than the original high-dimensional domain, LDMs substantially reduce computational cost while preserving expressive representational capacity. This design leads to remarkable performance in image synthesis and establishes a general architecture capable of learning highly nonlinear and structured mappings.

Building upon their success, a series of commercial-scale foundation models have adopted LDMs as core building blocks, achieving state-of-the-art results across a wide spectrum of applications, including video generation(Team et al., 2025; Ali et al., 2025), autonomous driving perception(Ni et al., 2025; Zheng et al., 2025), and robotic manipulation(Wen et al., 2025). These developments demonstrate that diffusion-based priors provide strong inductive biases that allow the models to capture multimodal, physically plausible, and semantically coherent structures.

Importantly, diffusion models exhibit an ability to handle otherwise intractable inverse or ill-posed problems by leveraging their robust generative priors. This capability makes them particularly promising for learning operators in sparse or partially observed PDE settings(Huang et al., 2024), where traditional machine learning architectures(Li et al., 2020b; Kovachki et al., 2023; Li et al., 2020a) and classical numerical solvers(Dhatt et al., 2012) often fail or become unstable. Thus, diffusion-driven operator learning provides a new direction for addressing complex PDE-related tasks under data scarcity and strong irregularity.

# B. Dataset Description

This section details the benchmark datasets employed in our experimental evaluations. We selected these datasets to cover a diverse range of partial differential equations (PDEs), addressing both forward and inverse problems. They provide a rigorous and standardized basis for evaluating the predictive accuracy and generalization capabilities of our proposed framework against established PDE solvers.

### B.1. Classic PDE Datasets

To systematically evaluate the performance of our model, we benchmark against several established methods using the datasets detailed below. Note that the variables $a$ and $u$ in the subsequent problem descriptions correspond to $A$ and $U$ in Eqs. (1), (2), and (3).

**Darcy Flow** The Darcy flow equations describe fluid dynamics within porous media under steady-state conditions. Our experiments utilize a static formulation of the Darcy flow problem, subject to homogeneous Dirichlet (no-slip) boundary conditions on the domain boundary $\partial\Omega$:

$$-\nabla \cdot (a(p)\nabla u(p)) = q(p), \quad p \in \Omega,$$
$$u(p) = 0, \quad p \in \partial\Omega.$$

Here, the function $a(p)$ represents the heterogeneous permeability field taking binary values, and $q(p) = 1$ is a constant forcing term. The corresponding PDE guidance function is formulated as $F(p) = \nabla \cdot (a(p)\nabla u(p)) + q(p)$.

**Inhomogeneous Helmholtz Equation** To model wave scattering in media with varying properties, we incorporate the static inhomogeneous Helmholtz equation, defined with homogeneous Dirichlet boundaries:

$$\nabla^2 u(p) + k^2 u(p) = a(p), \quad p \in \Omega,$$
$$u(p) = 0, \quad p \in \partial\Omega.$$

In this context, $a(p)$ serves as a piecewise constant source term, and $k$ denotes a real-valued constant. (Note that reducing $k$ to zero recovers the standard Poisson equation). For our experiments, the wavenumber is fixed at $k = 1$. The associated PDE guidance is therefore: $F(p) = \nabla^2 u(p) + k^2 u(p) - a(p)$.

**Unbounded Navier-Stokes Equation** We simulate the dynamics of an unbounded, incompressible fluid using the vorticity-velocity formulation of the Navier-Stokes equations:

$$\partial_\tau w(p,\tau) + v(p,\tau) \cdot \nabla w(p,\tau) = \nu \Delta w(p,\tau) + q(p), \ \ p \in \Omega, \ \tau \in (0,T],$$

$$\nabla \cdot v(p,\tau) = 0, \quad p \in \Omega, \ \tau \in [0,T].$$

Here, $w = \nabla \times v$ defines the scalar vorticity, $v(p,\tau)$ is the velocity vector field, and $q(p)$ acts as the external forcing. The kinematic viscosity is set to $\nu = 10^{-3}$, corresponding to a Reynolds number of $\mathrm{Re} = 1000$. Our model aims to learn the joint distribution linking the initial state $w_0$ and the state $w_T$ at time $T = 10$ (equivalent to one physical second). Because the extended temporal horizon makes exact computation of the full PDE residual computationally prohibitive, we leverage the vector calculus identity $\nabla \cdot (\nabla \times v) = 0$ to construct a simplified guidance function: $F(p,\tau) = \nabla \cdot w(p,\tau)$.

**Bounded Navier-Stokes Equation** Finally, we evaluate our framework on the 2D incompressible Navier-Stokes equations within a bounded physical domain, formulated using primitive variables (velocity $v$ and pressure $p$):

$$\partial_\tau v(p,\tau) + v(p,\tau) \cdot \nabla v(p,\tau) + \frac{1}{\rho}\nabla p = \nu \nabla^2 v(p,\tau), \ \ p \in \Omega, \ \tau \in (0,T],$$

$$\nabla \cdot v(p,\tau) = 0, \quad p \in \Omega, \ \tau \in (0,T].$$

For these simulations, we set $\nu = 0.001$ and assume a constant fluid density of $\rho = 1.0$. The geometry includes randomly sized cylindrical obstacles, with a turbulent inflow driven from the upper boundary. No-slip conditions ($v = 0$) are strictly enforced on the lateral walls ($\partial\Omega_{\text{left}}, \partial\Omega_{\text{right}}$) and the boundaries of the internal cylinders ($\partial\Omega_{\text{cylinder}}$). The objective is to capture the joint distribution between $v_0$ and $v_T$ over a timeframe of $T = 4$ (0.4 physical seconds). Consistent with the unbounded scenario, we utilize the divergence-free condition to define the PDE guidance function as $F(p,\tau) = \nabla \cdot v(p,\tau)$.

## C. GINO Architecture

In this section, we present the architectural design of GINO (Li et al., 2023), which serves as the foundational model for the subsequent analyses in this paper.

**GINO Encoder**  Following the problem setup 3.1, we consider a set of $M$ spatial coordinates $\mathbf{x}_m \subset \Omega$, where $m \in \{0, 1, \ldots, M\}$. At each location $\mathbf{x}_m$, a solution vector $\mathbf{u}(\mathbf{x}_m) \in \mathbb{R}^h$ is defined. This framework allows for the representation of solutions across arbitrary points in space.

To facilitate modeling on a uniform spatial grid, we map the input coordinates $\mathbf{x}_m$ and corresponding solutions $\mathbf{u}(\mathbf{x}_m)$ onto a set of uniformly sampled grid points $\mathbf{x}_l \subset \Omega$ for $l \in \{0, 1, \ldots, M_l\}$, with associated latent vectors $\mathbf{q}(\mathbf{x}_l) \in \mathbb{R}^h$ defined at each grid location. This mapping is performed via the following kernel-based integral operator.

$$\mathbf{q}(\mathbf{x}_l) = \int_\Omega \kappa(\mathbf{x}_l, \mathbf{x})\, \mathbf{u}(\mathbf{x})\, d\mathbf{x}. \tag{18}$$

In practice, following the approach (Li et al., 2023), the integration domain is restricted to a spatial ball of radius $r$, centered at $\mathbf{x}_l$, denoted as $\mathcal{B}_r(\mathbf{x}_l)$. This locality constraint ensures that each latent vector depends only on nearby solution values and spatial coordinates.

The kernel function $\kappa$ is parameterized by a neural network that takes as input a physical coordinate $\mathbf{x}$ and a latent coordinate $\mathbf{x}_l$, and outputs a scalar kernel value.

To compute the integral efficiently, it is approximated by a Riemann sum over $M_b < M$ physical coordinates $\mathbf{y}_b \subset \Omega$ that lie within the ball $\mathcal{B}_r(\mathbf{y}_l)$, where $b \in \{0, 1, \ldots, M_b\}$. This yields the following approximation:

$$\begin{aligned}
\mathbf{q}(\mathbf{x}_l) &= \int_{\mathcal{B}_r(\mathbf{x}_l)} \kappa(\mathbf{x}_l, \mathbf{x})\, \mathbf{u}(\mathbf{x})\, d\mathbf{x} \\
&\approx \sum_{b=1}^{M_b} \kappa(\mathbf{x}_l, \mathbf{x}_b)\, \mathbf{u}(\mathbf{x}_b)\, \mu(\mathbf{x}_b).
\end{aligned} \tag{19}$$

Here, $\mu(\mathbf{x}_b)$ denotes the Riemann weight associated with each physical point $\mathbf{x}_b$ within the spatial ball $\mathcal{B}_r(\mathbf{x}_l)$. This kernel integral effectively aggregates neighboring physical solutions for each latent location, and acts as an approximation of a spatial operator mapping the input field $\mathbf{u}(\mathbf{x})$ to the latent representation $\mathbf{q}(\mathbf{x}_l)$.

**GINO Decoder**  To map from the latent spatial grid to the physical domain, the kernel integral can be applied in reverse. Specifically, given a decoded latent representation $\mathbf{q}_d(\mathbf{x}_l)$ defined on the latent grid, the corresponding decoded physical solution $\mathbf{u}_d(\mathbf{x})$ can be computed via a Riemann sum:

$$\mathbf{u}_d(\mathbf{x}) = \sum_{b=1}^{M_b} \kappa(\mathbf{x}, \mathbf{x}_b)\, \mathbf{q}_d(\mathbf{x}_b)\, \mu(\mathbf{x}_b). \tag{20}$$

In this case, the summation is taken over the spatial ball $\mathcal{B}_r(\mathbf{x})$, and the coordinates $\mathbf{x}_b$ denote the latent grid points that lie within this ball.

This reverse aggregation process gathers information from neighboring latent vectors to reconstruct the solution at each physical location, while maintaining the property of discretization-invariance. Notably, this allows the latent representation to be decoded onto arbitrary spatial meshes by evaluating the Riemann sum at any query point $\mathbf{x}$.

# D. Pseudocode of Contrastive Learning

The pseudocode outlining the contrastive learning algorithm is provided in Algorithm 3, offering a step-by-step description of the training procedure.

# E. Experiments Information

We conducted our experiments within an open source framework *diffusers* (von Platen et al., 2022) in huggingface. All experiments were run on 8 Nvidia RTX4090 GPUs with 24GB memory.

---

**Algorithm 3** Pseudocode of Contrastive Learning

---

**Input**: Dataloader D ,
Full observations $U, A$

1: **repeat**
2:  Select minibatch $U_., A_.$ from dataloader D.
3:  Down-Sample $P_{.A}, P_{.U}$ from $A_., U_.$ .
4:  $C_{.A} = \text{SparseEncoderA}(P_{.A})$
    $\# [N, M, f_a] \rightarrow [N, h]$
5:  $C_{.U} = \text{SparseEncoderU}(P_{.U})$
    $\# [N, M, f_u] \rightarrow [N, h]$
6:  $K_{.A} = \text{GridEncoderA}(A_.)$
    $\# [W, W, f_a] \rightarrow [N, h]$
7:  $K_{.U} = \text{GridEncoderU}(U_.)$
    $\# [W, W, f_u] \rightarrow [N, h]$
8:  L2-Normalize $C_{.A}, C_{.U}, K_{.A}, K_{.U}$
9:  logitsF = $C_{.A} \times K_{.U}^T$                                        $\# [N, N]$
10:  logitsI = $C_{.U} \times K_{.A}^T$                                        $\# [N, N]$
11:  labels = range(N)
12:  Loss = Cross-Entropy-Loss(logitsF,labels,axis = 1)
13:  Loss += Cross-Entropy-Loss(logitsF,labels,axis = 0)
14:  Loss += Cross-Entropy-Loss(logitsI,labels,axis = 1)
15:  Loss += Cross-Entropy-Loss(logitsI,labels,axis = 0)
16:  Loss.backward()
17:  Update parameters
18: **until** converged

---

# F. (Hyper-) Parameter Information

See github project.

# G. Additional Experiment Results

*Table 3.* Relative $\ell_2$ errorRelative $\ell_2$ error on various observation rate

| Dataset | Task | 1% | 3% | 10% |
|---------|------|------|------|------|
| Darcy Flow | Fwd | 0.025 | 0.021 | 0.020 |
|  | Inv | 0.019 | 0.013 | 0.012 |
| Poisson | Fwd | 0.045 | 0.038 | 0.035 |
|  | Inv | 0.145 | 0.123 | 0.117 |

