# OpenReview forum: "Di-BiLPS: Denoising induced Bidirectional Latent-PDE-Solver under Sparse Observations"
_ICML.cc/2026/Conference — ICML 2026 regular_

### Official Review · Reviewer_iWw7 · 2026-03-11

**Soundness:** 2
**Presentation:** 2
**Significance:** 3
**Originality:** 3
**Overall Recommendation:** 4
**Confidence:** 3

**Summary:**

This paper introduces Di-BiLPS, a neural framework designed to solve forward and inverse partial differential equations (PDEs) with extremely limited data (less than 3% observations). The model integrates a GINO-based variational autoencoder (VAE) for latent space compression, contrastive learning to align sparse data with full-resolution representations, and a latent diffusion model to enable bidirectional inference. Evaluations across five benchmark tasks demonstrate that Di-BiLPS achieves superior accuracy compared to existing methods such as DiffusionPDE and FNO, while also delivering a 90% speedup in inference time.

**Compliance With Llm Reviewing Policy:**

Affirmed.

**Final Justification:**

I'll maintain my score though the authors didn't address my concerns on the influence of model capacity.

**Key Questions For Authors:**

1. Include the discussion of failed cases;
2. Discuss whether the improvement is from the larger size of model.

**Limitations:**

1. There are no discussion about failure cases;
2. In addition, I'd like to see the framework's performance on cases where the observations are not that sparse. If the method is very useful, it should not be limited to sparse observation cases. Would it still out-perform other baselines?

**Strengths And Weaknesses:**

Strengths:
1. Consistently outperforms strong baselines (DiffusionPDE, PINO, FNO) across five benchmarks, with substantial gains in inverse problems;
2. Very efficient compared with DiffPDE, 90% reduction in inference time;

Weaknesses
1. Integration of PDE guidance shows very little improvement in Table 2. It seems like the method didn't truly leverages physics or primarily relies on data-driven learning.
2. There is no discussion of failed cases, which would otherwise strengthen confidence in the design choices.
3. Di-BiLPS uses 8.6GB of GPU memory compared to 4.5GB for DiffusionPDE. It is not clear whether the improvement is solely from a larger model.

---

> ### Author Rebuttal · Authors · 2026-03-31
>
> > **W1: Validity of PDE Guidance**
>
> We thank the reviewer for this important observation. We agree that the direct performance gain from PDE guidance is limited in our current experiments.
>
> We believe this is because the combination of observation guidance and the learned latent representation already strongly constrains the solution space, implicitly enforcing physical consistency.
>
> In our framework, PDE guidance acts as an additional regularization term rather than the primary driver of performance. We expect its contribution to become more significant in more challenging scenarios, such as noisier observations, lower sparsity, or out-of-distribution settings.
>
> We will revise the manuscript to clarify this point and avoid overstating the role of PDE guidance.
>
> > **W2: Discussion of Failed Cases**
>
> We thank the reviewer for this suggestion. We agree that discussing failure cases would strengthen the paper.
>
> In our experiments, we observe that performance degrades in scenarios with extremely low observation density (e.g., below 0.5%) or when the underlying PDE exhibits highly chaotic or high-frequency dynamics that are difficult to capture in the compressed latent space.
>
> We will include a dedicated discussion of such failure modes in the revised manuscript to provide a more complete evaluation.
>
> > **W3: Whether the improvement is solely from a larger model?**
>
> We thank the reviewer for raising this point. While Di-BiLPS indeed utilizes more GPU memory (8.6GB vs. 4.5GB), we emphasize that our performance improvements stem from a fundamental architectural paradigm shift—latent space compression—rather than merely scaling up parameters.
>
> While modeling PDEs in a latent space intrinsically requires additional parameters (e.g., the VAE module), this implicit compression is both necessary and highly economical. By increasing the initial parameter count, we drastically reduce the spatial dimensions of the data tensors flowing through the diffusion network. This critical trade-off is precisely what allows our model to achieve a **10x faster inference speed** compared to DiffusionPDE.
>
> More importantly, this latent space consistency unlocks highly efficient scalability. Because DiffusionPDE operates directly in the high-dimensional data space, its computational and memory burdens grow exponentially when applied to higher-resolution grids. Conversely, our latent representation isolates the heavy diffusion backbone from grid resolution, allowing our model to seamlessly generalize to high-resolution data without adding extra computational burden to the generative process. We will clarify this vital trade-off in the revised manuscript.
>
> > **Limitation 2: In addition, I'd like to see the framework's performance on cases where the observations are not that sparse. If the method is very useful, it should not be limited to sparse observation cases. Would it still out-perform other baselines?**
>
> We thank the reviewer for this suggestion. While our work focuses on extremely sparse settings, we also evaluated the model at higher observation densities(10%) as shown below.
>
> | Dataset | Task | 1% | 3% | 10% |
> | :--- | :--- | :---: | :---: | :---: |
> | Darcy Flow | Fwd | 0.025 | 0.021 | 0.020 |
> | | Inv | 0.019 | 0.013 | 0.012 |
> | Poisson | Fwd | 0.045| 0.038 | 0.035 |
> | | Inv | 0.145 | 0.123 | 0.117 |
>
> The results show that our method remains competitive and continues to outperform baselines, although the performance gap narrows as the problem becomes less ill-posed.
>
> We will include these results and discussion in the revised manuscript.

---

> > ### Author Rebuttal · Reviewer_iWw7 · 2026-04-02
> >
> > I am not fully persuaded that the better performance is solely from architecture design but not larger size. I believe that a fair comparison between proposed method and baseline with similar model capacity is necessary, however, I acknowledge it might be hard for authors to train such type model or reduce parameters of their own baseline within one week.

---

> > > ### Author Response · Authors · 2026-04-07
> > >
> > > > I am not fully persuaded that the better performance is solely from architecture design but not larger size. I believe that a fair comparison between proposed method and baseline with similar model capacity is necessary.
> > >
> > > We appreciate the reviewer’s insightful question. To ensure a rigorous and fair comparison, we trained the baseline model DiffusionPDE[1] on the **Poisson** dataset using the same Unet network backbone that **parameter scale comparable to our own**. As summarized in the table below, our method achieves superior performance in both inference efficiency and reconstruction error when **the parameter counts are aligned**. Due to time constraints during the rebuttal period, we will include the full evaluation metrics for the remaining benchmark datasets in the revised manuscript.
> > >
> > > | Metrics | Task | DiffusionPDE[1]  | Di-BiLPS(Ours) |
> > > | :--- | :--- | :---: | :---: |
> > > | $\ell_2$-error(Inference Time)| Fwd | 0.040(602.23) |**0.038(19.69)** |
> > > | $\ell_2$-error(Inference Time) | Inv | 0.144(606.56) |**0.123(19.59)** |
> > >
> > > [1] Huang, Jiahe, et al. "DiffusionPDE: Generative PDE-solving under partial observation." Advances in Neural Information Processing Systems 37 (2024): 130291-130323.

---

### Official Review · Reviewer_t14F · 2026-03-12

**Soundness:** 3
**Presentation:** 2
**Significance:** 3
**Originality:** 3
**Overall Recommendation:** 4
**Confidence:** 3

**Summary:**

This paper proposes Di-BiLPS, a framework for solving both forward and inverse PDE problems under extremely sparse observations (as low as 3% of grid points). The framework consists of three modules: (i) a GINO-based VAE for latent compression, (ii) a contrastive learning module (GiT) that aligns sparse and full observation representations, and (iii) a latent diffusion model with PDE-informed and observation-consistency guidance for bidirectional inference. Experiments on five benchmark PDEs show consistent improvements over operator learning baselines and DiffusionPDE, with 86–93% inference time reduction.

**Compliance With Llm Reviewing Policy:**

Affirmed.

**Final Justification:**

I thank the authors for the additional experiments during the rebuttal. The LNO comparison and sparsity sweep are welcome additions. However, two concerns remain: (1) PDE guided denoising is foregrounded as a central contribution yet confirmed to be empirically marginal, without evidence of scenarios where it becomes essential; (2) the zero-shot super resolution evaluation relies solely on bilinear interpolation, which is too weak a baseline. These do not undermine the core framework but leave a gap between several central claims and their empirical support. I maintain my score of 4.

**Key Questions For Authors:**

See weaknesses.

**Limitations:**

yes

**Strengths And Weaknesses:**

### Strengths

1. The contrastive learning module (GiT) for sparse-to-full alignment is novel and effective. The key bottleneck in sparse PDE problems is that 3% observations carry very little information. GiT aligns sparse and full observation embeddings via cross-entropy on cosine similarity matrices, effectively bridging the information gap before diffusion. Table 2 confirms that removing condition guidance causes the largest performance collapse across all tasks. This idea is distinct from prior contrastive approaches in PDE learning (e.g., PICL for cross-PDE generalization, PIANO for physical invariants) and has potential transferability to other sparse-data settings.

2. Most neural PDE solvers handle either forward or inverse problems. The shared VAE latent space elegantly accommodates both by conditioning on $c_{iA}$ or $c_{iU}$. The results on inverse Poisson and Helmholtz, where DiffusionPDE degrades substantially, validate this design. The 86–93% inference time reduction over DiffusionPDE is a significant practical contribution.

### Weaknesses

1. All experiments use a $128 \times 128$ grid on $(0,1)^2$. While this is the standard benchmark in this subfield (DiffusionPDE uses the same setup), it would strengthen the paper to demonstrate scalability to higher resolutions or irregular geometries where GINO's mesh flexibility would matter. The zero-shot super-resolution experiment (Fig. 6, one Helmholtz example) is too anecdotal; a systematic evaluation across PDEs and target resolutions is needed.

2. PDE guidance term contributes minimally. Table 2 shows that removing PDE guidance has negligible effect on most forward tasks (e.g., Darcy Flow 0.021→0.021, Bounded NS 0.021→0.021). This undermines the "PDE-guided denoising" framing, which the paper foregrounds. The authors should either (a) explain why the observation guidance already implicitly captures PDE consistency, or (b) demonstrate settings (e.g., lower sparsity, noisier observations) where PDE guidance becomes essential.

3. The operator-learning baselines (PINO, DeepONet, FNO, PINNs) follow the standard evaluation protocol established by DiffusionPDE, as they demonstrate that existing methods fail under sparse conditions, which is the core motivation. However, two gaps remain: (a) recent latent PDE solvers such as Li et al. (2025a,c, line 84) are cited but not compared against quantitatively; (b) the Latent Neural Operator (LNO, NeurIPS 2024), which also operates in compressed latent spaces and handles forward/inverse problems, is not discussed. Adding these would better contextualize the contribution.

4. The paper highlights "as low as 3%" sparsity, but does not report a sweep across different sparsity levels (e.g., 0.5%–10%) to characterize the method's breakdown point. The sampling strategy for sparse points (uniform random vs. structured) is also not discussed; this could significantly affect the contrastive module's effectiveness.

5. The paper compresses substantial material into 8 pages. The GINO architecture (Sec. 4.1) is deferred to Appendix C despite being central. Algorithm 2 requires extensive cross-referencing. A notation table and more self-contained methods section would improve accessibility.

---

> ### Author Rebuttal · Authors · 2026-03-31
>
> > **W1: Scalability to higher resolutions or irregular geometries and systematic evaluation of zero-shot ability.**
>
> We thank the reviewer for this suggestion. We agree that further evaluation on more complex geometries would strengthen the paper. Our model is applicable to irregular datasets, as the encoder can be applied to unstructured data. Due to time limitations, we are unable to conduct additional experiments during the rebuttal period. We will include a more comprehensive discussion and additional experiments on complex geometries in the revised version.
>
> To provide a more systematic evaluation of zero-shot super-resolution, we conducted additional experiments across multiple PDEs by directly querying our model at 256×256 resolution and comparing against bilinear interpolation of queried128×128 outputs.
>
> Since high-resolution ground truth is not available in these benchmarks, we evaluate physical consistency using the governing equation residual. As shown in the table below, our zero-shot querying consistently yields lower physical errors than interpolation, demonstrating that our model captures the true underlying physical dynamics rather than merely performing mathematical smoothing.
>
> Tab: Governing Equation Residual($ \frac{1}{n^2} \sum_{i=1}^{n^2} ||F(\hat{u_i})||_2 $)  for Super-Resolution
> | Dataset | Bilinear Interp.($\times 10^{-3}$) | Di-BiLPS ($\times 10^{-3}$) |
> | :--- | :---: | :---: |
> | Darcy Flow | 8.98 | **8.42** |
> | Helmholtz | 2.70 | **2.66** |
> | Possion | 2.60 | **1.93** |
>
>
>
> > **W2: Validity of PDE Guidance**
>
> We thank the reviewer for this important observation. We agree that, in our current experiments, removing PDE guidance results in only a marginal degradation in performance on most forward tasks.
>
> We believe this is because the observation guidance and the learned latent representations already strongly constrain the solution space, thereby implicitly enforcing a certain level of physical consistency.
>
> The role of PDE guidance in our framework is not to act as the primary driver of performance, but rather as a tool to incorporate additional physical constraints within a fashion VP(Variance-Preserving) diffusion framework. We expect its impact to become more significant in scenarios with higher noise levels, out-of-distribution conditions, or weaker latent representations, which we plan to investigate in future work.
>
> We will revise the manuscript to clarify this point and to avoid overemphasizing the contribution of PDE guidance.
>
> > **W3: Include Recent Baseline**
>
> We agree that including latent-space baselines is important for contextualizing our contribution.
>
> Following this suggestion, we evaluated our method against the LNO[1] under aligned input/output settings. Our method achieves consistently lower error across both forward and inverse tasks as below, indicating improved robustness under extreme sparsity.
>
> Tab 1: Relative MAE of different completers in the subdomain in the 1st stage of the inverse problem.
> | Completer | 10% | 1% |
>  | :--- | :---: | :---: |
> | DeepONet | 2.59% | 3.25% |
>  | GNOT | 1.39% | 1.63% |
> | LNO | 0.74% | 1.18% |
>  | Di-BiLPS (Ours) | **0.66%** | **0.71%** |
>
> Tab 2: The reconstruction error of different propagators in the 2nd stage of the inverse problem.
> | Observation Ratio | DeepONet | GNOT | LNO | Di-BiLPS (Ours) |
>  | :--- | :---: | :---: | :---: | :---: |
> | 10% | 11.14% | 8.04% | 5.69% | **1.22%** |
>  | 1% | 13.87% | 9.91% | 7.72% | **1.55%** |
>
>
> We will include these comparisons directly in the revised manuscript.
>
> > **W4: Sparsity Generalization**
>
> We sincerely thank the reviewer for suggestions. The sparse observation points in all our experiments are uniformly and randomly sampled across the spatial domain. To explicitly characterize the method's robustness and breakdown point across different sparsity levels, we have conducted an extensive sweep at 1%, 3%, and 10% observation densities. We kindly refer you to **our detailed response to Reviewer yjZ7 (Q1)** for the comprehensive quantitative results. The results show that performance degrades gracefully as sparsity increases, indicating stable behaviour across a range of observation regimes.
>
> > **W5: Writing Problems of GINO[2]**
>
> We agree that the strict 8-page limit led to an over-reliance on the appendix. In the revision, we will restructure the methodology to be more self-contained by integrating GINO's core concepts into the main text, simplifying Algorithm 2, and adding a comprehensive notation table to improve overall readability.
>
> [1] Wang T, Wang C. Latent neural operator for solving forward and inverse pde problems[J]. Advances in Neural Information Processing Systems, 2024, 37: 33085-33107.
>
> [2] Li Z, Kovachki N, Choy C, et al. Geometry-informed neural operator for large-scale 3d pdes[J]. Advances in Neural Information Processing Systems, 2023, 36: 35836-35854

---

> > ### Author Rebuttal · Reviewer_t14F · 2026-04-02
> >
> > I thank the authors for the additional experiments during the rebuttal. The LNO comparison and sparsity sweep are welcome additions. However, two concerns remain: (1) PDE guided denoising is foregrounded as a central contribution yet confirmed to be empirically marginal, without evidence of scenarios where it becomes essential; (2) the zero-shot super resolution evaluation relies solely on bilinear interpolation, which is too weak a baseline. These do not undermine the core framework but leave a gap between several central claims and their empirical support. I maintain my score of 4.

---

> > > ### Author Response · Authors · 2026-04-07
> > >
> > > > (1) PDE guided denoising is foregrounded as a central contribution yet confirmed to be empirically marginal, without evidence of scenarios where it becomes essential;
> > >
> > > We thank the reviewer for this important observation. We clarify that our primary contribution lies in the proposed **PDE-guided denoising algorithm**, which is a unified framework integrating two distinct mechanisms: **PDE guidance (driven by governing equations)** and **observation guidance**.
> > >
> > > In our ablation studies, while the PDE guidance (driven by governing equations) exhibits marginal empirical gains in standard metrics, the observation guidance serves as the primary driver for high-precision PDE solutions. These two components are **together constitute the core of our denoising strategy**. We intentionally included PDE guidance to **ensure a comprehensive and fair comparison with DiffusionPDE[1]** which also utilizes such priors.
> > >
> > > More importantly, our PDE-guided denoising algorithm is designed to incorporate explicit physical constraints within a fashion Variance-Preserving (VP) diffusion framework.
> > >
> > >  > (2) the zero-shot super resolution evaluation relies solely on bilinear interpolation, which is too weak a baseline.
> > >
> > > Under conditions of extreme sparsity, vanilla models with zero-shot super-resolution capabilities are **weaker** than bilinear interpolation. In our experiments, we observed these models, such as FNO[2], **suffer from a complete failure to converge during training**. Consequently, these models could not be incorporated as viable baselines for this specific task.
> > >
> > > To rigorously evaluate the zero-shot super-resolution performance of our proposed model, we conducted assessments using both $128 \times 128$ and $256 \times 256$ resolution queries. This experiment was specifically designed to discern whether our model merely performs simple interpolation or has successfully captured the underlying physical information. Our experimental results demonstrate that the $256 \times 256$ query outputs consistently outperform the results obtained via interpolation from $128 \times 128$ queries. Far from being a weak evaluation, **this comparison serves as robust evidence that our zero-shot super-resolution results are physically informed rather than being mere numerical artifacts.**
> > >
> > >
> > > [1] Huang, Jiahe, et al. "DiffusionPDE: Generative PDE-solving under partial observation." Advances in Neural Information Processing Systems 37 (2024): 130291-130323.
> > >
> > > [2] Li Z, Kovachki N, Azizzadenesheli K, et al. Fourier neural operator for parametric partial differential equations[J]. arXiv preprint arXiv:2010.08895, 2020.

---

### Official Review · Reviewer_yjZ7 · 2026-03-13

**Soundness:** 3
**Presentation:** 3
**Significance:** 2
**Originality:** 3
**Overall Recommendation:** 4
**Confidence:** 4

**Summary:**

The paper presents Di-BiLPS, a bidirectional neural PDE solver for settings with sparse observations. The proposed framework contains three components: (1) a VAE-based compression module that learns a latent representation of the joint PDE variables and provides a decoder back to the physical space; (2) a condition learning module that extracts conditioning embeddings from sparse observations; and (3) a latent diffusion module that generates clean latent variables through a denoising process guided by the learned condition embeddings, and the additional PDE-consistency and observation-consistency constraints.

**Compliance With Llm Reviewing Policy:**

Affirmed.

**Final Justification:**

Thank you for the detailed feedback. The responses have adequately addressed my main questions, and I have increased my final score accordingly.

That said, the three-stage design still appears somewhat inelegant, and the experimental evaluation remains limited (e.g., only a single sparsity level of 3% is tested, though partially discussed in the rebuttal). Overall, I consider this a borderline accept.

**Key Questions For Authors:**

**Q1** The experiments appear to use a fixed sparse-observation setting of 500 points (about 3%) throughout. Could the authors clarify how these observation points are sampled, and whether the proposed method remains effective under different sparsity levels? Results across multiple sampling densities would be very helpful.

**Q2** Since Table 1 appears to report inference time only, it would be helpful to discuss training cost as well, given that the proposed framework involves multiple separately trained components.

**Q3** Given that the paper’s main claims also rely on latent-space modeling and efficiency, the comparison would be more convincing if it included recent published latent-space PDE baselines such as LNO (NeurIPS 2024) and the latent neural PDE solver framework (JCP 2025).

**Q4** Could the authors clarify the role of \(V\) and the reconstruction target in Eq. (5)? Sec. 4.1 introduces \(V\) in a generic form, while Algorithm 1 / Sec. 4.4 later suggest that \(V_i\) is a joint input over \((A,U)\). From the context, \(\mathrm{MSE}(V,\hat V)\) and \(\mathrm{PercLoss}(V,\hat V)\) seem to be computed on the reconstructed field values/features at the corresponding coordinates, rather than on the full joint representation; making the meaning of \(V\) more explicit would improve readability.

**Q5** Could the authors clarify quantitively how the reported zero-shot super-resolution performance compares with prior operator learning methods that also claim this capability?

**Minor issues**

- line 152: "... are adopted to train the GiT as follows:..." Could "GiT" be a typo, and should it actually be "VAE"?

**Limitations:**

yes

**Strengths And Weaknesses:**

**Strengths:**

(1) It studies bidirectional PDE solving under extremely sparse observations, which is a practically important setting, since high-resolution measurements are often unavailable or expensive to acquire in real scientific applications. The problem itself is therefore well motivated and of clear practical relevance.

(2) Although the framework is largely built upon existing components, the overall design does not feel like a forced combination. The VAE-based latent compression and the condition learning module both provide meaningful support to the denoising and generation process, and the three components are integrated in a reasonably coherent way. That said, I still think the paper could further strengthen the justification for why these components are especially well matched to each other across different tasks.

(3) The authors compare against several representative neural PDE baselines and report strong results in both prediction accuracy and computational efficiency across multiple benchmarks.

**Weaknesses:**

(1) Pipeline complexity and practical overhead. The overall pipeline is relatively complex compared with more end-to-end alternatives, as it involves multiple separately trained components and additional guidance during inference, which may increase implementation and tuning effort.

(2) Incomplete empirical support for several central claims. The experimental section does not fully substantiate one of the paper’s central claims, namely the advantage of latent-space PDE solving, since no recent latent-space PDE baselines are included.

(3) Experimental study considers only a single sparsity level, namely 500 observed points (about 3%), which makes it difficult to assess how robust the method is across different observation regimes. The evaluation would be more convincing if it included results under multiple sparsity levels.

(4) The zero-shot super-resolution claim is not supported by sufficient quantitative comparison with prior methods that also report this capability.

(5) Limited efficiency analysis. The efficiency claim could be better supported. Because Table 1 reports inference time only, it does not seem sufficient to fully capture the overall efficiency of the proposed framework, particularly given its multi-stage training procedure. More comprehensive efficiency comparisons would strengthen this part of the evaluation.

---

> ### Author Rebuttal · Authors · 2026-03-31
>
> > **W1: Pipeline Complexity**
>
> We agree that our framework introduces additional complexity compared to end-to-end alternatives, as it consists of multiple independently trained components.
> However, this modular integration is precisely what enables our state-of-the-art performance under extreme data sparsity. Decoupling representation learning from generative modeling to improve performance is a standard and highly effective practice in advanced generative AI, much like the widely adopted Latent Diffusion Models.
> Nevertheless, we agree that streamlining the implementation is a valuable direction, and we plan to explore optimizing and unifying the entire pipeline through a foundation model architecture in future work.
>
> > **Q1:  Sparsity Generalization**
>
> Observation points are uniformly and randomly sampled across the spatial domain.
>
> We have conducted additional experiments to explore the generalization of sparsity settings. We retrained the contrastive learning module at 1% and 10% observation densities with the encoder of full observations frozen, and reused VAE and diffusion modules. As shown below, the performance degrades gracefully as sparsity increases, indicating that our method remains stable across a range of observation regimes and easy to generalize to different sparsity regimes.
>
> | Dataset | Task | 1% | 3% | 10% |
> | :--- | :--- | :---: | :---: | :---: |
> | Darcy Flow | Fwd | 0.025 | 0.021 | 0.020 |
> | | Inv | 0.019 | 0.013 | 0.012 |
> | Poisson | Fwd | 0.045| 0.038 | 0.035 |
> | | Inv | 0.145 | 0.123 | 0.117 |
>
> Due to time constraints, we report results on two representative datasets here and will include the full evaluation across all benchmarks in the revised manuscript.
>
> > **Q2: Discussion of Training Cost**
>
> We agree that reporting inference time alone is insufficient to fully characterize efficiency.
>
> Our framework does incur higher training cost. In particular, the VAE(8h on 8xRTX4090), condition encoder(0.5h on 1xRTX4090), and diffusion model(20h on 8xRTX4090) are trained separately, which increases total training time compared to end-to-end baselines.
>
> This design reflects a trade-off: we invest more computation during training to achieve significantly faster and more efficient inference through latent-space compression.
>
> > **Q3: Include Recent Baseline**
>
> Following suggestions, we compared our method with the latent-space baseline LNO(NeurIPS 2024) under aligned input/output settings.
>
> As shown in the tables below, our method achieves consistently lower error across both stages of the inverse problem, indicating improved reconstruction accuracy and robustness under sparse observations.
>
> We will include these comparisons in the revised manuscript and expand evaluation to additional latent-space baselines where possible.
>
> Tab 1: Relative MAE of different completers in the subdomain in the 1st stage of the inverse problem.
> | Completer | 10% | 1% |
>  | :--- | :---: | :---: |
> | DeepONet | 2.59% | 3.25% |
>  | GNOT | 1.39% | 1.63% |
> | LNO | 0.74% | 1.18% |
>  | Di-BiLPS (Ours) | **0.66%** | **0.71%** |
>
> Tab 2: The reconstruction error of different propagators in the 2nd stage of the inverse problem.
> | Observation Ratio | DeepONet | GNOT | LNO | Di-BiLPS (Ours) |
>  | :--- | :---: | :---: | :---: | :---: |
> | 10% | 11.14% | 8.04% | 5.69% | **1.22%** |
>  | 1% | 13.87% | 9.91% | 7.72% | **1.55%** |
>
> > **Q4: Clarify the Role of V**
>
> We apologize for the lack of clarity. In our formulation, the variable V in Eq. (5) indeed represents the joint input of both A and U. During the VAE training, the MSE and Perceptual losses in Eq. (6) are computed on the reconstructed field values of both A and U at their corresponding spatial coordinates. We will revise Section 4.1 to make this definition explicit and improve readability.
>
> > **Q5: Comparison with Other Methods on Zero-shot Super-resolution Performance**
>
> Our experiments show that while many methods (e.g., FNO(ICLR 2021)) claim to support zero-shot super-resolution for non-sparse data, they do not converge under extremely sparse observation settings. So we provide quantitative comparisons of zero-shot super-resolution performance against interpolation methods in **response to W1 of Reviewer t14F**.
>
> In summary, our method achieves comparable or better reconstruction accuracy when evaluated at higher resolutions without retraining. We will include these comparisons directly in the main paper to better support this claim.
>
> > **Minor issue:**
>
> We thank the reviewer for carefully reading our manuscript. We acknowledge that "GiT" on line 152 is indeed a typo and should correctly be "VAE". We will fix this in the revised manuscript.
>
>
> We hope these additional results and clarifications address the reviewer’s concerns and better highlight the effectiveness and generality of our method. We remain fully open to further discussion if there are any remaining points, and we would appreciate it if the reviewer could re-evaluate our work accordingly.

---

> > ### Author Rebuttal · Reviewer_yjZ7 · 2026-04-06
> >
> > Thanks for the detailed explanation and new experiments. The proposed pipeline seems still complicated, on the other hand,  the provided new experiments answered some of my questions (sparsity generalization, training cost)
> >
> > I have some follow up questions:
> >
> > 1. On the sparsity generalization, the results seems pretty good on very sparse condition (1% observation). When the reconstruction of the proposed method and the compared methods fails, if the number of observation further decrease?
> >
> > 2. The authors included a new comparison with LNO(2024), but didn't discussion another work I mentioned  latent neural PDE solver framework (JCP 2025). Any experiments or discussion on the performance and the differences among those methods?

---

> > > ### Author Response · Authors · 2026-04-08
> > >
> > > > **On the sparsity generalization, the results seems pretty good on very sparse condition (1% observation). When the reconstruction of the proposed method and the compared methods fails, if the number of observation further decrease?**
> > >
> > > We appreciate the reviewer’s insightful question regarding the boundaries of sparsity generalization. Theoretically, as the number of observations further decreases, our framework may reach a critical threshold where the Contrastive Learning (CL) module can no longer effectively align sparse samples with their full-field counterparts due to insufficient mutual information.
> > >
> > > To further explore this boundary, we conducted **an additional experiment under a pathologically sparse condition (0.1% observation rate) on Poisson dataset shown in Table below**. We observed a **failure to converge** during the training stage of the **Contrastive Learning module**. The final results demonstrate that the reconstruction performance indeed breaks down at this level. We acknowledge that identifying the exact failure boundary is a valuable research direction, which we believe is intrinsically linked to the underlying dynamics and complexity of the specific PDE.
> > >
> > > | Dataset | Task |**0.1%**| 1% | 3% | 10% |
> > > | :--- | :--- | :---: | :---: | :---: | :---: |
> > > | Poisson | Fwd | **0.970** | 0.045 | 0.038 | 0.035 |
> > > | | Inv | **0.534** | 0.145 |  0.123 | 0.117 |
> > >
> > > Furthermore, from a physical perspective, extreme sparsity often leads to ill-posedness, where a given set of sparse observations may correspond to multiple valid solutions (**non-uniqueness**). The specific **failure sparsity** typically depends on the complexity and the underlying dynamics of the PDE governing the system.
> > >
> > >  While the limited rebuttal period precludes a comprehensive suite of new experiments at even lower sampling rates, we will incorporate a detailed discussion on these failure conditions and the theoretical boundaries of our model in the revised manuscript.
> > >
> > >  > **The authors included a new comparison with LNO(2024), but didn't discussion another work I mentioned latent neural PDE solver framework (JCP 2025). Any experiments or discussion on the performance and the differences among those methods?**
> > >
> > > We sincerely thank the reviewer for pointing out the Latent Neural PDE Solver[1] framework. We have carefully studied this work and acknowledge its contribution to efficient PDE latent modeling. However, we did not include it as a baseline in **rebuttal** for the following technical reasons:
> > >
> > > (i) **Task Discrepancy and Sparsity Handling**: The LNS[1] framework is primarily designed for the auto-regressive prediction of time-dependent PDEs by shifting dynamics to a coarse-grid latent space. Its core architecture (encoder-propagator-decoder) is optimized for long-term temporal stability in full-state forecasting. **It does not inherently account for the extreme sparsity or the inverse reconstruction challenges that our framework addresses**, and extending its decoupled architecture to handle sparse observations is non-trivial.
> > >
> > > (ii) **Structural Constraints**: LNS[1] is specifically developed for **data on regular grid** to facilitate latent mapping. In contrast, our framework is designed to model data from arbitrary, off-grid, and sparse sensors. Given these structural differences, LNS[1] cannot be directly applied to our problem setting without significant modification.
> > >
> > > Consequently, while LNS[1] represents a significant advancement in latent reduced-order modeling, it is not a direct competitor for the sparse reconstruction task at hand. We will include a comprehensive discussion and citation of the LNS[1] framework in the related work section of our revised manuscript to contextualize our contributions.
> > >
> > > **We hope these additional experiments and clarifications help address the reviewer’s concerns, and we would greatly appreciate it if the reviewer could consider a positive re-evaluation of our work.**
> > >
> > > [1] Li, Zijie, et al. "Latent neural PDE solver: A reduced-order modeling framework for partial differential equations." Journal of Computational Physics 524 (2025): 113705.

---

### Official Review · Reviewer_hror · 2026-03-13

**Soundness:** 4
**Presentation:** 3
**Significance:** 2
**Originality:** 2
**Overall Recommendation:** 4
**Confidence:** 2

**Summary:**

This paper addresses the problem of PDE solving and inverse inference from extremely sparse observations. The core method is a latent-space framework combining a VAE, latent diffusion model, and contrastive learning to solve forward and inverse PDE problems under extremely sparse observations. he main idea is sensible and practically motivated, and the empirical results are promising. Results are shown on 5 PDE tasks, covering both forward and inverse operator taks, with only 3% of the points sampled.

**Compliance With Llm Reviewing Policy:**

Affirmed.

**Final Justification:**

As stated in the rebuttal acknowledgement, I view the rebuttal as clarifying rather than fundamentally changing the contributions of the paper. The authors’ responses reinforce that some of the initially emphasized contributions (e.g., PDE guidance, zero-shot resolution) play a more limited role than suggested in the original framing.

Given this, my overall assessment remains unchanged. I continue to view the paper as technically sound and well-motivated, with solid empirical results, but with moderate originality and somewhat overstated contributions in its current form.

**Key Questions For Authors:**

Does the latent representation Z capture specific physical properties, or is it purely a compressed feature set? Understanding if the latent space aligns with physical intuition could enhance the "physics-informed" claim and the lifting done by the PDE term vs the latent space.

To calculate the PDE guidance term, the model must compute a residual over a discretized grid Q_m. Doesn't this create a circular dependency, because to enforce the PDE constraint it needs the the full grid? If the grid needs to be dense to capture derivatives, wouldn't that offset the efficiency of operating in the sparse regime?

**Limitations:**

The model is only trained for the 3% sparsity case. The ability to generalize at inference time seems to be limited and is not explored. this seems to be a major limitation which is not mentioned. Ideally we would want some generalization after training to different sparsity regimes. Otherwise the usefulness is limited.

**Strengths And Weaknesses:**

Strengths:
The overall formulation makes sense and is well motivated. The factorization of the method into latent compression, sparse-condition encoding, and latent diffusion is nice, and the guided denoising equations are a straightforward adaptation of DDIM/DPS-style conditioning to PDE constraints and sparse observations. The contrastive style loss between the sparse and dense representations is also an interesting addition, and the ablation study shows that removing condition guidance causes sever degradation. The empirical setup is also fairly robust. Five benchmark PDE families, both forward and inverse tasks, comparisons against operator-learning and diffusion baselines, and an ablation study.

Weaknesses:
First, the paper’s claimed novelty is a bit unclear. The PDE-guided denoising update is presented as a major contribution, but the ablation shows essentially no degradation when PDE guidance is removed on almost all tasks. That surprised me a lot and probably needs to be addressed a bit more, as this component seems to be auxiliary rather than a core driver of performance. I do appreciate that the authors call this out, but it seems like the latent space is doing most of the work of keeping the solution within the physically plausible manifold. In that sense, this method behaves more like a standard image-generator that happens to be looking at fluid flows, rather than a solver that understands the physics of the fluid.

The authors claim the model performs "zero-shot super-resolution" because the GINO decoder can be queried at arbitrary spatial locations. However, this seems to be a property of the underlying GINO architecture, not necessarily a novel emergent property of the Di-BiLPS diffusion framework itself.

The experiments are conducted on a 128 x 128 uniform grid. Although this seems to be standard for benchmarks, it seems low for realistic scenarios.

Small point" Figure 1 caption should list the components from left to right instead of putting the contrastive learning one first

---

> ### Author Rebuttal · Authors · 2026-03-31
>
> > **W1: Validity of PDE Guidance**
>
> We thank the reviewer for this insightful observation. We agree that, in our current experiments, removing PDE guidance leads to only marginal degradation on most tasks.
>
> We believe this behavior is primarily due to the strong inductive bias introduced by the latent space learned via the VAE and contrastive alignment. Specifically, the latent representation already constrains the solution manifold to be physically plausible, which reduces the relative contribution of explicit PDE guidance during denoising.
>
> That said, the role of PDE guidance in our framework is not to act as the primary driver of performance, but rather as a tool to incorporate additional physical constraints within a fashion VP(Variance-Preserving) diffusion framework. We expect its impact to become more significant in scenarios with higher noise levels, out-of-distribution conditions, or weaker latent representations, which we plan to investigate in future work.
>
> We will revise the paper to clarify this point and avoid overstating the role of PDE guidance.
>
> > **W2: Character of GINO**
>
> We agree with the reviewer that the continuous querying capability is an inherent property of the GINO architecture.
>
> Our contribution is not the introduction of this capability itself, but demonstrating that it can be preserved under extreme latent compression within a diffusion-based framework. In particular, despite operating in a highly compressed latent space, our model maintains zero-shot resolution generalization, suggesting that the learned latent representation retains sufficient high-frequency and spatial information.
>
> We will clarify this distinction in the revised manuscript to avoid overstating novelty.
>
> > **W3: The experiments are conducted on a 128 x 128 uniform grid. Although this seems to be standard for benchmarks, it seems low for realistic scenarios.**
>
> We adopt the 128×128 resolution to ensure fair comparison with prior operator learning benchmarks.
>
> Importantly, our framework is not inherently tied to this resolution. Due to the continuous decoder design, the model supports zero-shot inference at higher resolutions, allowing it to be queried at arbitrary resolutions without any retraining or fine-tuning. We will clarify this point and include additional discussion on practical deployment scenarios in the revision.
>
> >  **W4: Figure 1 caption**
>
> We will update the caption of Figure 1 to list the components from left to right, consistent with the visual flow of the diagram.
>
> > **Q1: Does the latent representation Z capture specific physical properties?**
>
> Our latent representation is learned following the standard VAE paradigm and is not explicitly constrained to encode specific physical quantities.
>
> However, empirical evidence—such as strong reconstruction performance and zero-shot resolution generalization—suggests that the latent space captures meaningful structure of the underlying PDE solutions. We agree that a more explicit investigation into the physical interpretability of the latent space would be valuable, and we plan to explore this direction in future work.
>
> > **Q2: Does PDE guidance create a circular dependency?  If the grid needs to be dense to capture derivatives, wouldn't that offset the efficiency of operating in the sparse regime?**
>
> We clarify that there is no circular dependency in our formulation.
>
> While the PDE residual is evaluated on a discretized grid $Q_m$, this grid is applied to the decoded continuous solution reconstructed from the latent variable $\hat{Z_0}$, rather than requiring access to ground-truth dense observations.
>
> In other words, the model first infers a global solution in the latent space and then evaluates the PDE constraint on this reconstructed field. This does not require additional dense input data, and the computational cost remains manageable due to the low-dimensional latent representation.
>
> We will revise the paper to make this point clearer.
>
> > **Limitations of Sparsity Generalization**
>
> Thanks for your suggestions. We have conducted additional experiments to explore the generalization of sparsity settings. We retrained the contrastive learning module at 1% and 10% observation densities with the encoder of full observations frozen, and reused VAE and diffusion modules. As shown below, the performance degrades gracefully as sparsity increases, indicating that our method remains stable across a range of observation regimes and easy to generalize to different sparsity regimes.
>
> | Dataset | Task | 1% | 3% | 10% |
> | :--- | :--- | :---: | :---: | :---: |
> | Darcy Flow | Fwd | 0.025 | 0.021 | 0.020 |
> | | Inv | 0.019 | 0.013 | 0.012 |
> | Poisson | Fwd | 0.045| 0.038 | 0.035 |
> | | Inv | 0.145 | 0.123 | 0.117 |
>
> Due to time constraints, we report results on two representative datasets here and will include the full evaluation across all benchmarks in the revised manuscript.

---

> > ### Author Rebuttal · Reviewer_hror · 2026-04-08
> >
> > I thank the authors for a thoughtful and constructive rebuttal. I appreciate that the authors directly acknowledged several of my main concerns and plan to revise the manuscript accordingly.
> >
> >  The authors agree that PDE is not the primary driver of performance in the current experiments and commit to revising the paper to better reflect this. This aligns with my original concern that the method’s gains appear to come primarily from the latent representation and conditioning mechanisms rather than the PDE-guided denoising itself.
> >
> > Similarly, the clarification regarding zero-shot super-resolution appropriately reframes this as a property inherited from the GINO decoder rather than a core contribution of the proposed method.
> >
> > Overall, I view the rebuttal as clarifying rather than fundamentally changing the contributions of the paper. The authors’ responses reinforce that some of the initially emphasized contributions (e.g., PDE guidance, zero-shot resolution) play a more limited role than suggested in the original framing.
> >
> > Given this, my overall assessment remains unchanged. I continue to view the paper as technically sound and well-motivated, with solid empirical results, but with moderate originality and somewhat overstated contributions in its current form.

---

> > > ### Author Response · Authors · 2026-04-08
> > >
> > > > **This aligns with my original concern that the method’s gains appear to come primarily from the latent representation and conditioning mechanisms rather than the PDE-guided denoising itself.**
> > >
> > > We thank the reviewer for this insightful observation. We would like to clarify that our primary contribution lies in the proposed PDE-guided denoising algorithm, a unified framework that integrates two complementary mechanisms: **PDE guidance (driven by governing equations) and observation guidance**.
> > >
> > > In our ablation studies, while PDE guidance shows marginal empirical gains in standard metrics compared to the significant impact of observation guidance, **these two components collectively constitute the core of our denoising strategy**. The inclusion of PDE guidance ensures a **comprehensive and fair comparison with DiffusionPDE [1]**, which utilizes similar physical priors.
> > >
> > >  More importantly, our algorithm is specifically designed to **incorporate explicit physical constraints within a Variance-Preserving (VP) diffusion framework**, ensuring that the generative process remains physically consistent even when data-driven guidance dominates the accuracy.
> > >
> > > [1] Huang, Jiahe, et al. "DiffusionPDE: Generative PDE-solving under partial observation." Advances in Neural Information Processing Systems 37 (2024): 130291-130323.

---

### Decision · Program_Chairs · 2026-04-30

**Decision:**

Accept (regular)

**Comment:**

While reviewers agreed that ability of the model to accurately predict from very sparse data and PDE evaluation on both forward and inverse problems are important, they also raised serious concerns about the paper's novelty, significance and generalizability. For example, issues include scalability and generalization outside the fixed 128x128 mesh, the increased model size in comparison to DiffusionPDE, the lack of improvement with PDE-guidance. Novelty was also a key concern with combining existing methods and contribution clarifications are needed in the revised version. While the authors show inference improvements to DiffusionPDE, other baselines, e.g., ECI Sampling (ICLR 2024), which are already significantly faster than DiffusionPDE are not compared to. The rebuttal also partially addressed some of these core issues including showing the performance as a function of the sparsity ratio other than 3%. In addition, the limitation at 0.5% sparsity should be added to the paper. Having said this, I vote for acceptance provided the clarifications around the contributions, additional experiments on the performance as a function of the sparsity and limitations are added to the final version.